



# Monitoring Surface Water Dynamics in the Prairie Pothole Region Using Dual-Polarised Sentinel-1 SAR Time Series

Stefan Schlaffer[1,2], Marco Chini[3], Wouter Dorigo[1], and Simon Plank[2]

[1]Technische Universität Wien, Department of Geodesy and Geoinformation, Wiedner Hauptstr. 8-10, 1040 Vienna, Austria
[2]German Aerospace Center, Earth Observation Center, Münchener Str. 20, 82234 Wessling, Germany
[3]Luxembourg Institute of Science and Technology, 41 rue du Brill, 4422 Belvaux, Luxembourg

**Correspondence:** Stefan Schlaffer (stefan.schlaffer@geo.tuwien.ac.at)

**Abstract.** The North American Prairie Pothole Region (PPR) represents a large system of wetlands with great importance for biodiversity, water storage and flood management. Knowledge of seasonal and inter-annual surface water dynamics in the PPR is important for understanding the functionality of these wetland ecosystems and the changing degree of hydrologic connectivity between them. Optical sensors have been widely used to calibrate and validate hydrological models of wetland dynamics. Yet, they are often limited by their temporal resolution and cloud cover, especially in the case of flood events. Synthetic aperture radar (SAR) sensors, such as the ones on board the Copernicus Sentinel-1 mission, can potentially overcome such limitations. However, water extent retrieval from SAR data is often affected by environmental factors, such as wind on water surfaces. Hence, for reliably monitoring water extent over longer time periods robust retrieval methods are required.

The aim of this study was to develop a robust approach for classifying open water extent dynamics in the PPR and to analyse the obtained time series covering the entire available Sentinel-1 observation period from 2015 to 2020 in the light of ancillary data. Open water in prairie potholes was classified by fusing dual-polarised Sentinel-1 data and high-resolution topographical information using a Bayesian framework. The approach was tested for a study area in North Dakota. The resulting surface water maps were validated using high-resolution airborne optical imagery. For the observation period, the total water area, the number of water bodies and the median area per water body were computed. The validation of the retrieved water maps yielded producer's accuracies between 84% and 95% for calm days and between 74% and 88% on windy days. User's accuracies were above 98% in all cases, indicating a very low occurrence of false positives due to the constraints introduced by topographical information.

Surface water dynamics showed strong intra-annual dynamics especially in the case of small water bodies (< 1 ha). Water area and number of small water bodies decreased from spring throughout summer when evaporation rates in the PPR are typically high. Larger water bodies showed a more stable behaviour during most years. During the extremely wet period between the autumn of 2019 and mid-2020, however, the dynamics of both small and large water bodies changed markedly. While a larger number of small water bodies was encountered, which remained stable throughout the wet period, also the area of larger water bodies increased, partly due to merging of smaller adjacent water bodies. However, the area covered by small water bodies was more stable than the area covered by large water bodies. This suggests that large potholes released water faster via the drainage network, while small potholes released water mainly to the atmosphere via evaporation. The results





demonstrate the potential of Sentinel-1 data for high-resolution monitoring of prairie wetlands. Limitations exist related to wind inhibiting correct water extent retrieval and due to the rather low temporal resolution of 12 days over the PPR.

## 1 Introduction

Surface water dynamics in wetland ecosystems play an important role in water storage variability of a region (Acreman, 2012) and are of great importance for flood management (Huang et al., 2011b), biodiversity (Cohen et al., 2016), groundwater recharge (Mitsch and Gosselink, 2000) and biogeochemical cycles (Cheng and Basu, 2017). The distribution of wetlands of different sizes in time and space is a key factor determining their function in a landscape. While large wetlands store considerable water volumes (Liu and Schwartz, 2011), small wetlands fulfil important roles for biodiversity by acting as habitats (Krapu et al., 2018). The ecological functioning of wetlands depends greatly on the abundance of wetlands of different sizes in the landscape, which is subject to land-use changes and climate variability (McKenna et al., 2019).

The Prairie Pothole Region (PPR) of North America covers an area of over 780,000 km$^2$ and consists of millions of shallow depressions formed during glacier retreat at the end of the last glacial period. These depressions contain open water bodies whose areas vary between one square metre and several square kilometres. They can store considerable amounts of water during rainfall events, which contributes to flood mitigation in downstream populated areas (Huang et al., 2011b). The wetlands of the region are of great importance for the waterfowl population of North America (Mitsch and Gosselink, 2000). Many of the smaller wetlands fall dry during the summer months, especially in drier years, and re-fill during spring snowmelt or intense rainfall events (Huang et al., 2011a; Montgomery et al., 2018). The water body size distribution of the PPR and its variability have thus attracted considerable interest in the hydrologic community (Bertassello et al., 2019; Liu and Schwartz, 2011; Proulx et al., 2013). Van Meter and Basu (2015) revealed a preferential loss of smaller wetlands compared to historic data, which they attributed to the artificial draining of small, upland wetlands and to the preferential restoration of large, lowland wetlands. Furthermore, distances between individual wetlands as well as between wetlands and the river network have increased, with important implications for their connectivity, e.g., in terms of habitat function. In another study, distributions of pothole sizes from a hydrological model run over the twentieth century (Liu and Schwartz, 2011) were analysed with respect to their inter-annual and seasonal changes. The abundance of larger potholes ($> 1$ ha) was found to be relatively unaffected by short-term climatic variations, whereas the size distribution of smaller wetlands changed due to the typical seasonal cycle of snowmelt in spring and gradual drying-out due to high evaporation rates in summer.

Direct evidence to support such conceptual models of water surface dynamics in the PPR is mainly based on Earth observation (Rover and Mushet, 2015). Remote sensing-based approaches can be used to derive wetland extent over large areas at a comparatively low cost (Ozesmi and Bauer, 2002). The resulting wetland extent maps have been used to calibrate and validate hydrological models simulating wetlands behaviour (McIntyre et al., 2014). Both optical and synthetic aperture radar (SAR) data have been used for mapping surface water extent in the PPR (Proulx et al., 2013; Vanderhoof et al., 2016; Brooks et al., 2018; Bolanos et al., 2016). In addition to satellite data, very high-resolution (HR) airborne optical as well LiDAR data have been used both for estimating surface water extent and routing of water flow between potholes (Wu and Lane, 2017; Wu et al.,





2019; Vanderhoof and Lane, 2019). However, satellite-based imagery has received the bulk of attention to data, likely due to its
lower cost. Landsat imagery in particular has received much consideration due to its long time period covered, i.e., from 1972
to present. Vanderhoof et al. (2016) mapped inter-annual changes in wetland extent using time series of cloud-free Landsat
scenes at an approximately annual interval over a 21 year period. Wetland extent and connectivity were found to correlate
with each other as well as with climatological indices and runoff. The authors noted that their study was limited by the spatial
resolution of the Landsat data and by the fact that sub-annual dynamics in small wetlands, e.g., in response to rainfall events,
could not be assessed. In a study using eight Landsat scenes acquired over the course of two years, Brooks et al. (2018) found
variations in total water surface area of ca. 50% within a catchment. Although they analysed only a relatively small number
of images, their results highlight the dynamic nature of surface water extent in the PPR. The use of Landsat imagery for mon-
itoring surface water extent in the PPR is limited by its temporal resolution, which is additionally degraded by cloud cover,
and its relatively coarse spatial resolution of 30 m (Rover and Mushet, 2015). In this context, Vanderhoof and Lane (2019)
assessed the Landsat-based Global Surface Water (GSW) dataset (Pekel et al., 2016) for mapping the distribution of wetland
sizes in the PPR and characterising their interactions. The authors concluded that analysis of the Landsat-based product alone
would suggest that the landscape in the PPR is dominated by wetlands of sizes 0.2 ha to 8.0 ha. Using a dataset based on HR
imagery that was pan-sharpened to 0.5 m spatial resolution, however, resulted in smaller wetlands dominating the distribution
of wetland sizes. Based on this product they also detected narrow interactions between wetlands in the form of channels and
locations where adjacent wetlands merged during wet periods (Vanderhoof and Lane, 2019).

However, HR satellite or airborne imagery are typically not available at the very short time intervals necessary to resolve
intra-annual variations in water body sizes. Moreover, the flood mitigation potential of the wetlands in the PPR is a function
of the water volume existing at the beginning of a flood event (Huang et al., 2011b). While cloud cover additionally limits the
temporal resolution of optical data, SAR sensors could provide a more continuous monitoring of surface water extent. SAR
data have been successfully used for wetlands mapping (e.g. Brisco, 2015; Reschke et al., 2012; Schlaffer et al., 2016; White
et al., 2015). In addition to the ability of microwave radiation to penetrate clouds, SAR sensors are not only highly sensitive
to the occurrence of open water surfaces (Richards, 2009) but also to flooding beneath vegetation (Tsyganskaya et al., 2018).
Recent missions, such as Sentinel-1, also offer data at spatial resolutions comparable to or higher than Landsat and temporal
sampling intervals in the order of several days. Open water surfaces act as specular reflectors and, as a result, appear dark in
the resulting imagery (Giustarini et al., 2016). Factors, such as wind (Bartsch et al., 2012) or vegetation protruding through
the water surface, however, lead to an increase in the energy amount scattered back to the sensor and, hence, increase false
negative rates in the classification. Dry, sandy areas (Martinis et al., 2018), wet snow (Bartsch et al., 2012) or tarmac can
be confused with open water during SAR image classification. In recent years, several studies have aimed at mapping and
monitoring surface water dynamics in the PPR from SAR data. In particular, the analysis of multi-polarised data has received
attention, as co and cross-polarised data respond differently to scattering mechanisms like surface and volume scattering.
Bolanos et al. (2016) used dual-polarised Radarsat-2 time series to map open water dynamics in the Canadian PPR and applied
different thresholds to backscatter and image texture. Montgomery et al. (2018) analysed time series acquired by Radarsat-2 for
classifying prairie wetlands according to their hydroperiod, i.e., the number of days per year that a wetland is covered by water.





Strong fluctuations in water extent in accordance with precipitation inputs were reported especially for the more hydrologically
disconnected study sites. The rather long revisit cycle of Radarsat-2 of 24 days was mentioned as a major limiting factor for
characterising surface water dynamics. In a study by Huang et al. (2018), data acquired by Sentinel-1, which has a temporal
resolution of 12 days over most of the PPR, have been used to classify open water in the PPR of North Dakota. A set of
polarised indices was created from the dual-polarised imagery and used together with backscatter coefficients as features in
a random forest classifier trained on reference surface water products, such as the aforementioned GSW dataset. The authors
noted that limitations of their approach relate to omission of water-covered areas due to inundated vegetation and the spatial
resolution of the sensor as well as commission errors due to smooth surfaces resembling open water in SAR imagery (Huang
et al., 2018).

Such limitations, along with the short observation periods available yet for most SAR missions, in comparison to, e.g.,
Landsat, pose the greatest hindrances for a wider uptake of SAR data for a long-term monitoring of wetland dynamics. In
contrast to most SAR missions up to date, the two-satellite Sentinel-1 constellation focuses on providing consistent data over
longer time periods (Torres et al., 2012), which is ensured by the launch of Sentinel-1 C/D planned from 2022 onwards (ESA
CEOS EO Handbook, 2021). Therefore, there is a need for novel algorithms making use of the capabilities of Sentinel-1,
such as dual polarisations and its high spatial temporal resolutions, while addressing the abovementioned limitations, such as
misclassification due to water surfaces roughened by wind or land surfaces resembling open water. In the field of flood mapping,
the inclusion of ancillary topographic information has been used to minimise the influence of these factors. Such ancillary
information can be integrated into the classification workflow either by masking during post-processing (e.g. Westerhoff et al.,
2013; Schlaffer et al., 2015) or by probabilistic data fusion of SAR and topographic data (e.g. D'Addabbo et al., 2016).

Here, a retrieval algorithm for open water bodies in the PPR based on dual-polarisation Sentinel-1 data is proposed. We use
a probabilistic approach combining SAR backscatter and information derived from a LiDAR-based digital elevation model in
order to minimise the occurrence of false positives caused by bare areas, tarmac or wet snow, which has been identified as a
limiting factor in the aforementioned study (Huang et al., 2018). The method is applied to the full time series of Sentinel-1
imagery available for the snow-free months between 2015 and 2020. We hypothesise that a time series of water extent maps
at sub-monthly intervals will facilitate the analysis of both intra-annual and inter-annual variations in the distribution of water
body sizes. As mentioned earlier, modelling studies, such as Liu and Schwartz (2011), have demonstrated the sensitivity of
small water bodies to intra-annual variations, whereas larger water bodies were only affected by deluge or drought periods at
larger time scales. Our focus is both on inter-annual surface water dynamics as well as on the impacts of short, intense rainfall
or snowmelt events on the number of water bodies and the area covered by them. To our knowledge, this is the first time
that inter-annual wetland dynamics in the PPR are studied using the entire length of the Sentinel-1 time series. Furthermore,
this study represents the first analysis of wetland dynamics during the flood events of 2019, which caused large areas in the
Midwest to be inundated (Yin et al., 2020).





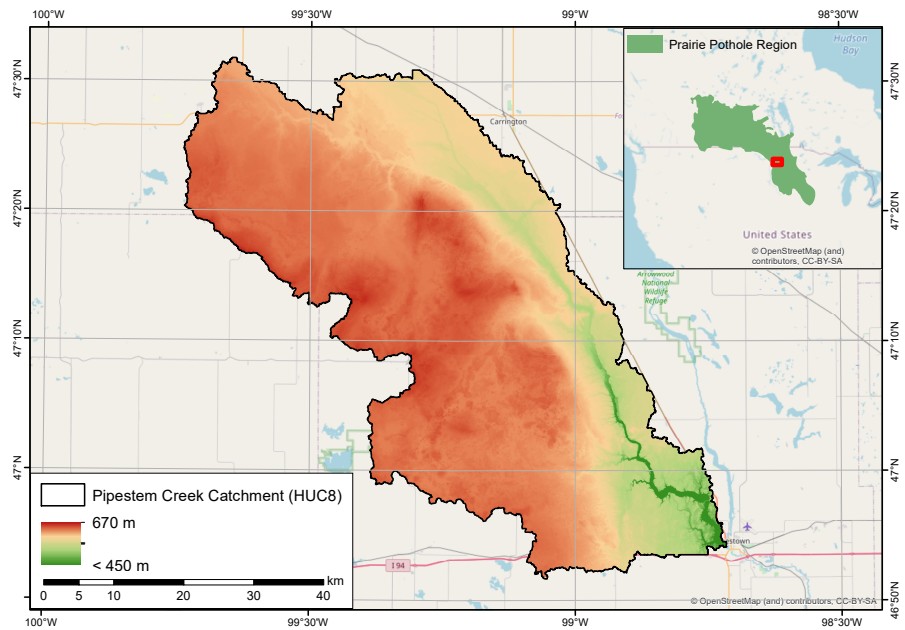

**Figure 1.** Pipestem Creek Catchment. The inset shows the location of the study area within the PPR. OpenStreetMap is used as background. The elevation data are provided by North Dakota State Water Commission (2018) under the Creative Commons license.

## 2   Material and methods

### 2.1   Study area

The study area comprises the Pipestem Creek catchment in North Dakota (ND), USA (Fig. 1). The catchment has an area of ca. 2,770 km$^2$ and forms part of the PPR, which covers a large part of the Great Plains in the Northern USA and Southern

Canada. Climate is continental with cold, dry winters (Wu and Lane, 2017) and a long-term average annual precipitation of approximately 440 mm (Huang et al., 2011a), most of which occurs during the summer months (Fig. 2a). Inter-annual precipitation variability is high: during the study period from 2015 to 2020, annual precipitation measured at Jamestown Regional Airport varied between 341 mm (in 2017) and 661 mm (in 2016). The study area is predominantly under agricultural use, while natural vegetation mainly consists of grassland. According to the National Agricultural Statistical Service (NASS)

Cropland Data Layer (CDL) for 2015 (US Department of Agriculture, 2016), dominant crop types in the area are corn, soybean, wheat and sunflowers.

Discharge in the Pipestem Creek shows large variability (Fig. 2b). Runoff peaks typically occur in spring as a result of snowmelt; however, their magnitude varies considerably, e.g., between the spring seasons of 2016 and 2017. In 2019, runoff peaks occurred in both spring and autumn. In that year, ND and several other Midwestern states documented their wettest

year on record. The high runoff led to widespread flooding in the Missouri, Arkansas and Mississippi river basins. In addition,





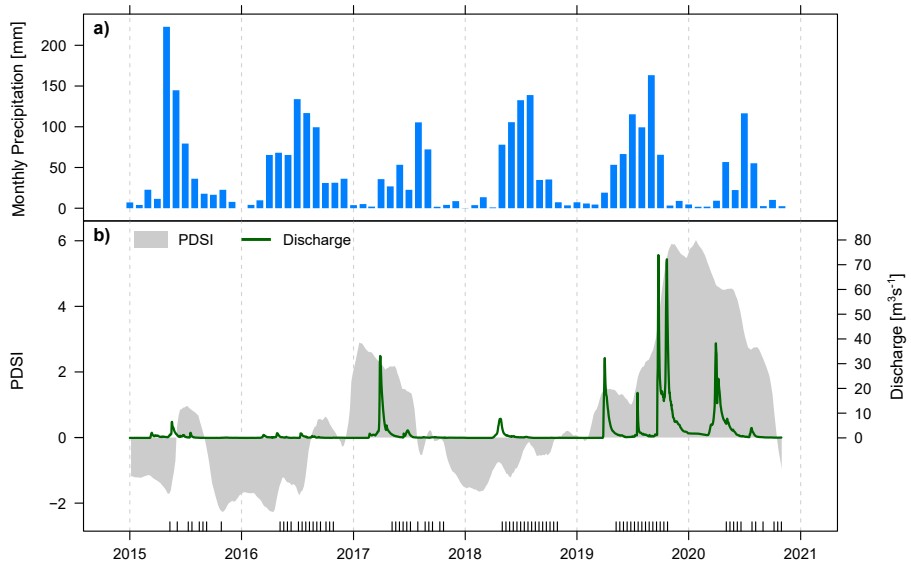

**Figure 2.** a) Monthly precipitation sums at Jamestown Regional Airport (from Global Summary of Day, ©National Climatic Data Center, NESDIS, NOAA, U.S. Department of Commerce); b) daily runoff at Pingree (©USGS) and GRIDMET-derived PDSI (Abatzoglou, 2013) during the observation period. Rug at the bottom marks the dates of the Sentinel-1 acquisitions used in this study.

blizzards in mid-October led to the declaration of a state-wide flood emergency (Umphlett, 2019). Between 2015 and 2018, the Palmer Drought Severity Index (PDSI), which was derived from GRIDMET data (Abatzoglou, 2013) and averaged over the study area, oscillated between -2 and +2 indicating normal to slightly dry and slightly wet conditions, respectively. In the last two years of the observation period, however, PDSI increased until reaching values $> 5$ in late autumn of 2019, indicating
extremely wet conditions which persisted until summer of 2020. The occurrence of discharge peaks in 2017, 2019 and 2020 coincided with periods of positive PDSI (Fig. 2b).

## 2.2 Data and pre-processing

### 2.2.1 Sentinel-1 data

Surface water dynamics were derived from a stack of Sentinel-1 Interferometric Wide Swath (IWS) imagesry. Sentinel-1A was
launched in April 2014, its twin satellite Sentinel-1B followed in April 2016. Both satellites carry a C-band SAR instrument operating at a wavelength of ca. 5.6 cm. The ground-range detected (GRD) product has a spatial resolution of ca. 20 m (Torres et al., 2012). Data for the study area is available from March 2015 onwards. Since wet snow and ice cover on lakes can alter backscatter behaviour, the study was limited to the months May to October, which we assumed to be mostly snow-free. A total number of 74 scenes acquired between May 2015 and October 2020 was downloaded from the Copernicus Open Access Hub.





From 2016, data were available at an interval of 12 days with a few exceptions (Fig. 2b). All the scenes used in this study
were acquired from the same relative orbit (number 34) and available in both vertical send – vertical receive (VV) and vertical
send – horizontal receive (VH) polarisations. The downloaded GRD scenes were filtered using a Gamma-MAP speckle filter
(Lopes et al., 1993) with a window size of $3 \times 3$ pixels and radiometrically calibrated to obtain the backscattering coefficient
$\sigma^0$. Terrain correction was carried out using the Range-Doppler approach and the digital elevation model from the Shuttle

Radar Topography Mission (Farr et al., 2007). The scenes were resampled to a common grid with a pixel spacing of 10 m
in the Universal Transverse Mercator (UTM Zone 14 North) projection. The SAR data were pre-processed using the Sentinel
Application Platform (SNAP version 7), provided by the European Space Agency (ESA).

### 2.2.2    Topographical data

A digital terrain model (DTM) with a resolution of 1 m was available from the North Dakota State Water Commission (2018)

(Fig. 1). The DTM is based on LiDAR data acquired in autumn 2011. The data package downloaded from the North Dakota
GIS Hub also included polygons of water bodies classified based on the LiDAR intensity data. For the comparison with
Sentinel-1 data, the DTM and the water bodies were aggregated to the 10 m grid mentioned above. The water body polygons
were aggregated by retaining pixels as water pixels if they were covered by a water polygon by more than 50%. Then, the
Height Above Nearest Drainage (HAND), $z_{\mathrm{HAND}}$, was computed from the DTM. HAND consists of the relative elevation of a

DTM pixel above the nearest pixel pertaining to the drainage network (Rennó et al., 2008) and has been used in flood remote
sensing for masking areas that are not prone to floods (Westerhoff et al., 2013; Schlaffer et al., 2015; Twele et al., 2016). In
order to take the special environmental conditions encountered in the PPR into account, we used both the identified potholes as
well as the drainage network obtained from the DTM after filling the pothole sinks as drainage pixels. The drainage network
was extracted using the *r.watershed* tool in GRASS GIS (GRASS Development Team, 2017).

### 2.2.3    Land use/land cover

As a source of information on land use/land cover, the US Department of Agriculture (USDA) NASS CDL for North Dakota
was downloaded (henceforth referred to as CDL). The CDL for 2015 is based on data from Landsat 8, DEIMOS-1 and UK2
that were acquired during that year. The CDL has a spatial resolution of 30 m and is provided under a Creative Commons
licence (US Department of Agriculture, 2016). The CDL was resampled to the same grid as the Sentinel-1 data using nearest

neighbour resampling.

### 2.2.4    Validation data

The surface water extents were validated for three different dates in 2016, 2017 and 2019 using aerial imagery from the National
Agriculture Imagery Program (NAIP). These years were selected to have a representation of dry and wet years. False-colour
composites of the images are shown in Appendix A (Fig. A1). NAIP imagery has a spatial resolution of 1 m and comprises

four bands in the visible and near-infrared (NIR) portions of the electromagnetic spectrum. For validating the water extents





derived from Sentinel-1, we sampled points across the extent of the NAIP images and classified them manually into water and non-water classes. Although wetlands occur frequently in the study area, a random sample would likely underrepresent surface water. To account for this class imbalance, a stratified random sampling approach was applied. With the help of the DTM-derived potholes, 200 pixels per class were randomly sampled from potholes and upland areas throughout each of the three NAIP footprints, resulting in 400 reference points for each reference image (shown in Fig. A1). The NIR band was especially useful in identifying areas where vegetation was emerging from water surfaces. We assumed that such conditions would fundamentally impact radar backscatter from these areas. Whenever vegetation was protruding through the water surface around a sampling point, the respective point was classified as non-water as the proposed approach applies only to open water surfaces.

## 2.3 Water extent delineation

Calm, open water surfaces typically cause specular reflection of incident microwave radiation. In SAR scenes with an approximately balanced mix of open water and land classes this phenomenon leads to bimodal grey-value histograms. However, these classes rarely occur at similar proportions within a scene and, as a result, bimodality of the histogram is not commonly observed. Efforts have been made to balance water and non-water classes by subsampling areas from SAR images where water and land pixels are equally represented (Martinis et al., 2009; Schlaffer et al., 2016; Chini et al., 2017). An example of such an effort are split-based approaches (Martinis et al., 2009; Chini et al., 2017), where image subsets showing a bimodal grey-value distribution are automatically selected from a SAR scene based on a set of pre-defined criteria. This approach is especially useful if the approximate locations of the water bodies or flooded areas are not known *a priori*.

The locations of prairie potholes are governed by the landscape features of the PPR and have been reported to be relatively stable over longer time periods (Bolanos et al., 2016). Therefore, we chose to treat the potholes as the baseline of the study. The pothole locations were determined based on the water bodies product contained in the 2011 LiDAR data. We assumed that this dataset contained more water bodies than what could be classified from satellite data due to the higher spatial resolution and the fact that the data had been acquired during extremely wet conditions (Wu and Lane, 2017). Therefore, it was regarded as a suitable baseline dataset for monitoring surface water dynamics in as many potholes as possible.

An overview of the water classification workflow is shown in Fig. 3. In contrast to other studies on remote sensing-based water retrieval, which treat a scene uniformly, i.e., by estimating the statistical distributions of water and non-water classes along with classification thresholds across the entire image, the class-specific backscatter distributions were estimated locally, i.e., for each of the known pothole locations. The reasoning behind this approach is that backscatter values may vary considerably over large regions over both water and land surfaces due to wind or semi-submerged vegetation, on the one hand, and variations in soil moisture, wet snow cover or vegetation structure and moisture, on the other.

In order to estimate the backscatter distribution to be expected for open water an independent reference layer is required (Schlaffer et al., 2017), e.g., derived from optical data. Here, we chose the CDL for 2015 as reference layer. Backscatter values for open water bodies delineated in the CDL were extracted for the months May to October, henceforth denoted $\sigma^0_{w,p}$ for each of the two polarisations $p \in \{VV, VH\}$. Pixels along the borders of water areas in the CDL were excluded to minimise border





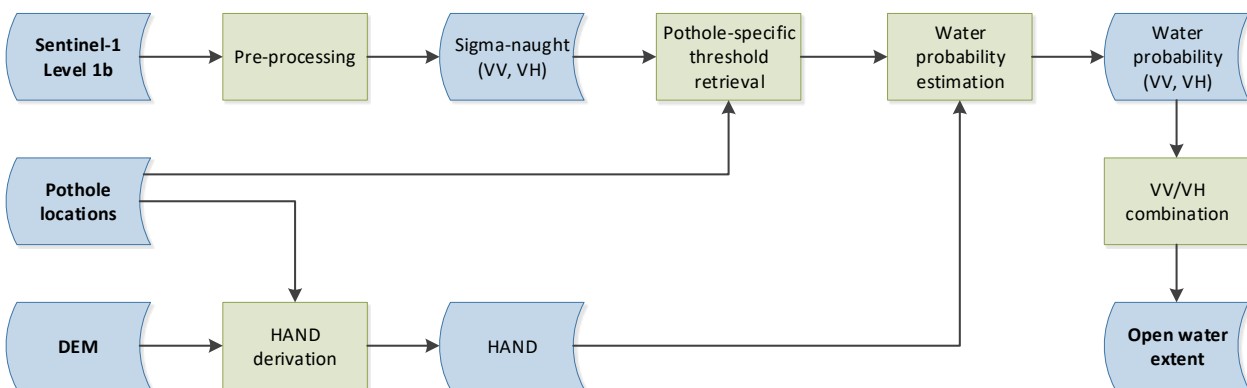

**Figure 3.** Sentinel-1 processing workflow for dynamic open water classification.

220 effects. The mean of $\sigma^0_{w,p}$ is $\bar{\sigma}^0_{w,p}$. For each of the previously delineated potholes, we checked if at least 10 pixels had $\sigma^0_p$ values lower than $\bar{\sigma}^0_{w,p}$. If this condition was not fulfilled it was assumed that no open water was present in the respective pothole. Otherwise, we proceeded to check the bimodality of the $\sigma^0$ distribution within the pothole. We followed the approach by Chini et al. (2017) for this. As a first guess, the histogram was automatically split using Otsu's approach (Otsu, 1979). The bimodality was then assessed using Ashman's $D$ (Ashman et al., 1994), which is defined as

$$D = \sqrt{2}\frac{|\mu_1 - \mu_2|}{\sigma_1^2 + \sigma_2^2}, \tag{1}$$

where $\mu_1$, $\mu_2$ are the means of the two histogram portions and $\sigma_1^2$, $\sigma_2^2$ their variances. A value of $D > 3$ was assumed to be an indication of a bimodal histogram. If this condition was not fulfilled, the region around the pothole was extended to include neighbouring pixels in order to include a higher number of non-water pixels and checked again for bimodality. For each pothole, the sampling region around the pothole was further extended by including neighbouring pixels up to a maximum number of

230 10 iterations. If a bimodal histogram was encountered before reaching this number of iterations the final Otsu threshold for the respective pothole was saved. In the following, the threshold value derived for $\sigma^0_p$ is denoted $\tau_p$. The parameters of the water and land distributions for each pothole were estimated as

$$\mu_{w,p} = \sum \sigma^0_{w,p}/N_{w,p}, \qquad \sigma^2_{w,p} = \frac{1}{N_{w,p} - 1}\sum \sigma^0_{w,p}, \tag{2}$$

and

$$\mu_{l,p} = \sum \sigma^0_{l,p}/N_{l,p}, \qquad \sigma^2_{l,p} = \frac{1}{N_{l,p} - 1}\sum \sigma^0_{l,p}, \tag{3}$$





respectively, where $\sigma_{w,p}^0$ are all $\sigma_p^0 \leq \tau_p$, $\sigma_{l,p}^0$ are all $\sigma_p^0 > \tau_p$. $N_{w,p}$ and $N_{l,p}$ are the number of $\sigma_{w,p}^0$ and $\sigma_{l,p}^0$ values, respectively. The probability of a pixel belonging to the water distribution given a certain value of $\sigma_p^0$ was computed as

$$p(W|\sigma_p^0) = \frac{p(\sigma_p^0|W)p(W)}{p(\sigma_p^0|W)p(W) + p(\sigma_p^0|L)[1-p(W)]}. \tag{4}$$

$p(\sigma_p^0|W)$ and $p(\sigma_p^0|L)$ are the probability density functions (PDFs)

$$p(\sigma_p^0|W) = \frac{1}{\sqrt{2\pi\sigma_{w,p}^2}}e^{\frac{\sigma_p^0 - \mu_{w,p}}{2\sigma_{w,p}^2}} \tag{5}$$

and

$$p(\sigma_p^0|L) = \frac{1}{\sqrt{2\pi\sigma_{l,p}^2}}e^{\frac{\sigma_p^0 - \mu_{l,p}}{2\sigma_{l,p}^2}} . \tag{6}$$

$p(W)$ in Eq. 4 denotes the prior probability of a pixel being land or water. If no information is available, $p(W) = 0.5$ can be used giving equal prior probability to the land and water distributions (Giustarini et al., 2016). However, as the location of the open water surfaces was known to be mainly confined by the topography of the potholes, which changes little over time (Bolanos et al., 2016), we chose to incorporate this knowledge. Bayes' theorem has been used as an efficient framework to fuse information from different sources, e.g., for obtaining SAR-based flood delineation maps (e.g. Frey et al., 2012; D'Addabbo et al., 2016; Li et al., 2019). Here, $p(W)$ was modelled using logistic regression with $z_{\mathrm{HAND}}$ as predictor:

$$p(W) = \frac{1}{1 + e^{-b_0 - b_1 z_{\mathrm{HAND}}}}, \tag{7}$$

where $b_0, b_1$ are regression parameters which were estimated using 20 samples, each consisting of 5000 land and 5000 water pixels identified from the CDL. An additional sample of 4985 land and 4668 water pixels not containing any of the pixels used for training was set aside for testing. When sampling the training and testing data, pixels in a buffer region along the border between land and water were excluded to obtain a pure sample of each class. Pixels that are identified as land in the CDL may still be prone to the occurrence of wetlands. For example, they may be located in depressions that have been drained or may have been not covered by water when the imagery used for the CDL has been acquired. In order to account for potential sampling biases Eq. (7) was fitted to each of the 20 training samples separately and the average of the estimated parameters $b_{0,i}, b_{1,i}, 0 < i \leq 20$ was used to estimate $p(W)$.

The estimated $p(W|\sigma_p^0)$ values were classified into water and non-water classes taking into account the spatial relationship between areas with high $p(W|\sigma_p^0)$, which changes over time, and the static pothole extents derived from the DTM. This approach served to minimise the occurrence of isolated clusters of water pixels far from potholes which may occur due to speckle or dry land surfaces appearing as dark areas in the SAR image. This condition was implemented using a region growing approach, using the DTM-based pothole area as seed. The region growing was limited to a maximum of 10 iterations to prevent "spilling" of positive pixels into large portions of the scene. The idea is illustrated in Fig. 4. Two potholes are separated in the





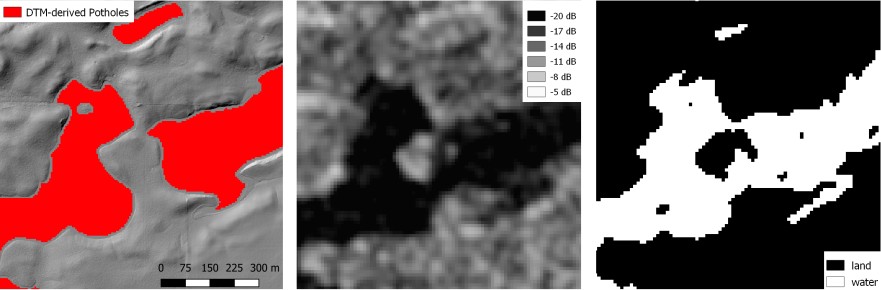

**Figure 4.** Left: DTM hillshade and derived potholes; centre: Sentinel-1 $\sigma_{VV}^0$ on 2017-05-31; right: derived open water extent.

DTM by a low topographic barrier (left). However, the potholes are visibly merged together in the Sentinel-1 image shown in
the centre. After the region growing, the classification result shows the merged water bodies (right).

In addition, the final water area should take into account the information from both VV and VH polarisations. The respective
water probabilities, $p(W|\sigma_{VV}^0)$ and $p(W|\sigma_{VH}^0)$, were combined based on their values. We opted for a rather conservative
approach for combining the two datasets to minimise the occurrence of false positives. In summary, we applied the following
conditions for classifying dynamic open water from water probability:

(i) $p(W|s0_{VV}) > 0.8 \vee p(W|s0_{VH}) > 0.8 \vee [p(W|s0_{VV}) > 0.5 \wedge p(W|s0_{VH}) > 0.5]$, where $\vee$ is a logical OR and $\wedge$ is a
logical AND,

(ii) $\leq 10$ pixels distance to original pothole area,

(iii) derived water area connected with original pothole area (taking account all eight neighbours of each pixel).

For validation, producer's and user's accuracies were computed for the water class using the combined product against the
reference data described in Section 2.2.4. Each of the three reference datasets was compared to the Sentinel-1-derived water
bodies closest in time to the date of the respective NAIP acquisition. In order to estimate the impact of using VV or VH
polarisation, accuracies were also computed for each polarisation separately using a threshold of $p(W|\sigma_p^0) > 0.5$.

## 2.4 Analysis of surface water dynamics

Prairie wetlands can merge over time with neighbouring wetlands or split into separate water bodies. Hence, monitoring the
area of individual water bodies is challenging. To track surface water dynamics across the study area, we computed the total
water area, areas covered by individual water bodies and the number of discrete water bodies for each of the observation dates.
We were furthermore interested in the inter and intra-annual dynamics of wetlands of different sizes. For this purpose, water
bodies were divided in four different size bins and the contribution of each bin to the total water extent was tracked along the
observation period. The metrics were computed using the R package *landscapemetrics* (Hesselbarth et al., 2019). In accordance
with the spatial resolution of the Sentinel-1 sensor, before deriving the metrics we applied a minimum mapping unit of 0.04 ha,





equal to the area of four pixels, to remove small clusters of water pixels from the result. Such small clusters are often the result of noise.

## 3 Results and Discussion

### 3.1 Open water classification

The obtained $p(W)$ values are shown in Fig. 5. It can be seen that high $p(W)$ predominantly occur in potholes and along rivers and streams, whereas most of the upland areas are assigned values close to zero. We validated the estimated $p(W)$ using an independent test sample with labels assigned from the LiDAR-based water map. Calculating an overall accuracy for the test sample would not take into account the unknown occurrence of potential wetlands in the CDL land class. Therefore, we computed the sensitivity of our approach as $s = TP/(TP+FN)$, where $TP$ denotes true positives and $FN$ false negative test

values, after classifying the predicted $p(W)$ into binary values using a threshold of $p(W) = 0.5$. The sensitivity quantifies the capability of the classifier to correctly identify existing water bodies and was calculated as 0.985.

Open water maps were produced for the months May to October from 2015 to 2020 (Fig. 6c,d). In Fig. 6c), most of the water bodies were identified in both VV and VH polarisations. However, the subset also shows several wetlands only detected in VV, visible as light-blue areas in the RGB composite. VH backscatter over land areas is mainly related to volume scattering. In areas

with sparse vegetation density, this may lead to a low contrast between water and non-water in the VH band of the image. In such cases, no suitable threshold could be determined. The subset shown in Fig. 6d) contains several wetlands only classified in the VH data (pink colour). Comparison with the corresponding backscatter image (Fig. 6b) reveals that VV backscatter seems to be increased over these water bodies as indicated by the reddish colours. This, in turn, again leads to lower contrast between water and surrounding non-water classes. In some cases, water bodies could not be detected in either polarisation,

e.g., the large water body in the centre of Fig. 6d,f). This is often the case when, on the one hand, ice cover is present or the water surface is roughened by wind and, on the other hand, when the surrounding land surface appears dark, as in the centre of Figure 6b). Such darker areas are often related to grassland and sparsely vegetated areas, which can be distinguished in the false-colour images in Fig. 6e,f) by their paler appearance from vegetated fields visible in bright red colour. Agricultural fields tend to appear brighter in the SAR imagery (Fig. 6a,b) owing to high soil moisture and vegetation influencing both VV and

VH-polarised backscatter.

Comparison with independent NAIP data, which was carried out for three dates, resulted in high producer's accuracies (> 84%) for two dates and very high user's accuracies for all three dates (> 98%), suggesting some underestimation of the water extent (Table 1). The high user's accuracies are the result of a very low number of false positives. The underestimation of the water extent is to be expected due to the difference in spatial resolution between Sentinel-1 (ca. 20 m) and the reference

data (1 m). In such cases, validation points located close to the edges of objects may coincide with mixed pixels in the lower resolution imagery. In the case of the validation carried out for July 2016, producer's accuracies were lower, especially for VV, possibly due to wind roughening the water surface. Such effects are visible in the NAIP imagery and may have occurred also on the following day when the Sentinel-1 scene was acquired. The extent of the NAIP image acquired in 2019 is located in the



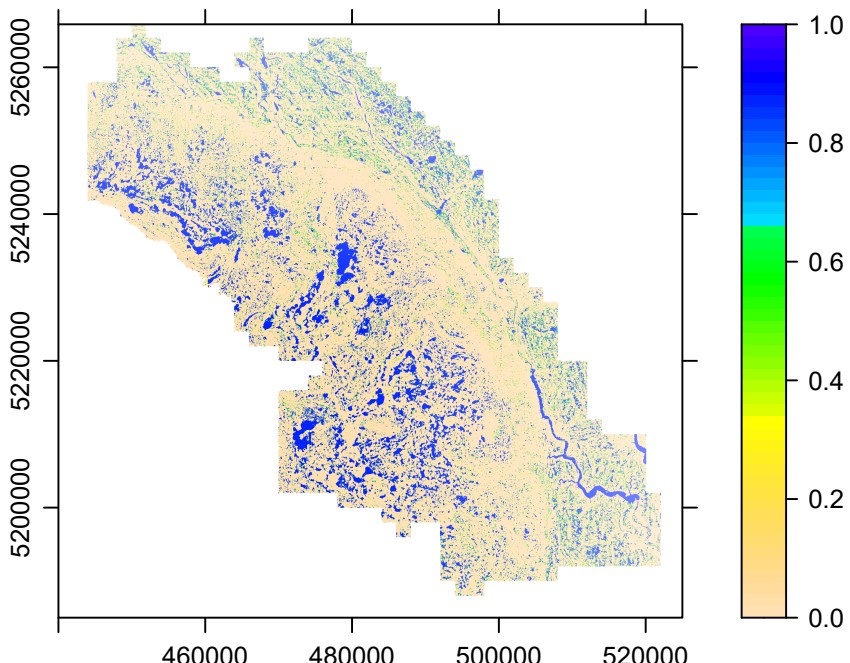

**Figure 5.** Map of predicted $p(W)$. Scales are in UTM (zone 14) coordinates.

upper Pipestem catchment and dominated by a high number of small water bodies which were sometimes not detected by our

approach, leading to somewhat lower producer's accuracies below 90%.

The accuracies reported here are similar to the ones reported by Huang et al. (2018) who compared open water extent derived from dual-pol Sentinel-1 data against reference extents derived from NAIP imagery over the Pipestem Basin. The authors of that study also used the scene acquired on 5 July 2016 for validation. For the water class, they obtained producer's accuracies between 64% and 92%, and user's accuracies between 87% and 99%. For the wind-affected scene of July 2016, a

producer's accuracy of 64% was reported, while here, 87.8% were obtained with the combination of VV and VH polarisations. The inclusion of topographic information may have helped to mitigate the effect of the backscatter increase due to the wind-roughened water surface. This interpretation is supported by the relatively high producer's accuracy for VV of 74.5% on that day. In comparison to Huang et al. (2018), user's accuracies reported here were consistently higher, suggesting a lower over-estimation of water extent. We attribute this to the integration of HAND in the Bayesian framework, which helped to constrain

water extent retrieval even in image areas where the contrast between water and non-water pixels was low. Such low contrast areas often lead to a "spilling" into non-water areas during the region-growing process.



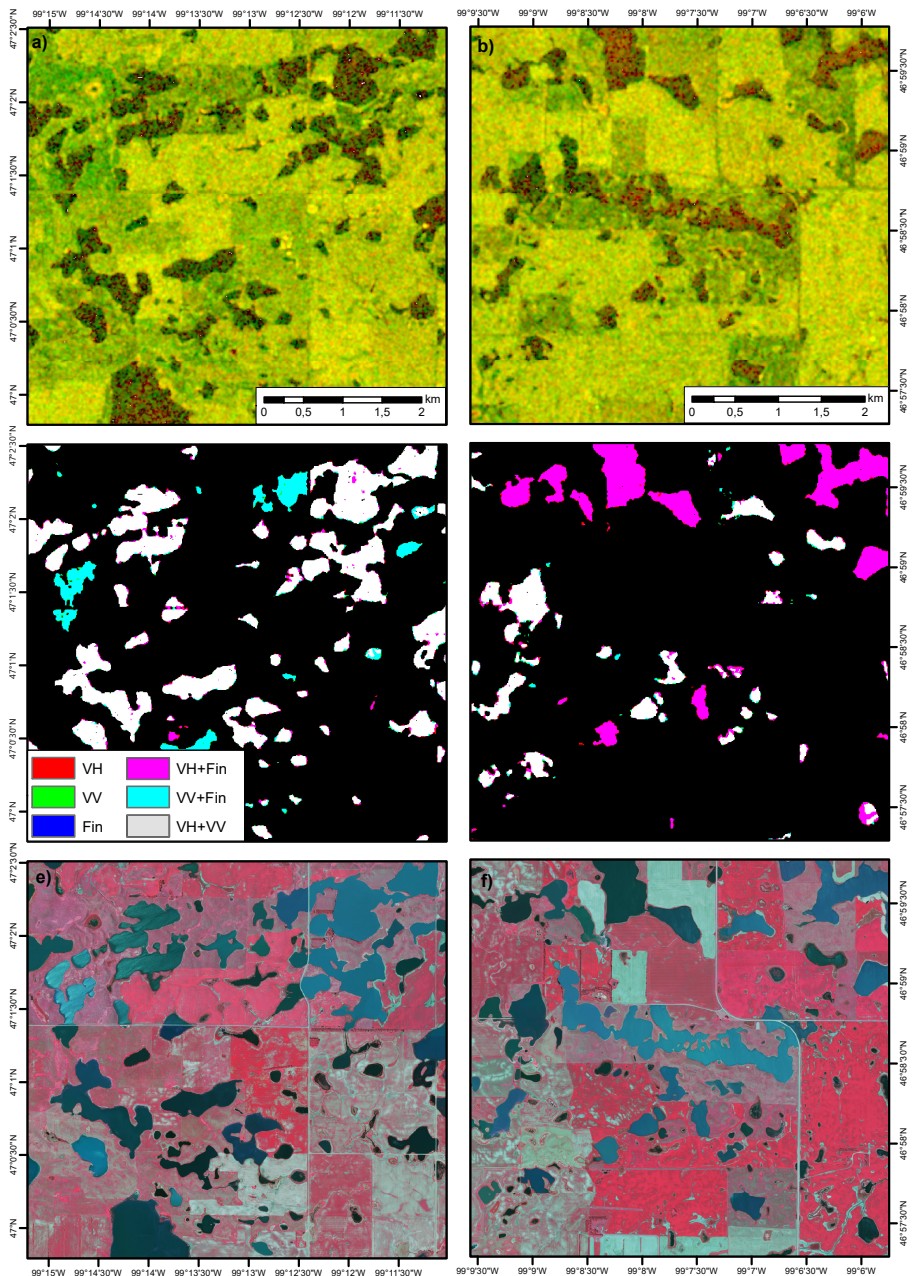

**Figure 6.** a,b) colour composites of $\sigma^0_{VV}$ (red) and $\sigma^0_{VH}$ (green) on 5 July 2016; c,d) classification results as RGB composite - - red: VH, green: VV, blue: Final classification (Fin); e,f) false-colour composites of NAIP imagery acquired 4 July 2016.

Wind is an important environmental factor in the prairies and has been shown to affect SAR water extent retrieval by causing scattering from roughened water surfaces (Bartsch et al., 2012). As a result, the contrast between water and non-water areas





**Table 1.** Accuracy values of open water classification for three dates.

| Date | Reference | Polarisation | Producer's accuracy | User's accuracy |
|---|---|---|---|---|
| | | VV | 74.5 | 99.3 |
| 2016-07-05 | 2016-07-04 | VH | 87.2 | 99.4 |
| | | VV+VH | 87.8 | 99.4 |
| | | VV | 91.7 | 100.0 |
| 2017-08-23 | 2017-08-27 | VH | 92.8 | 98.8 |
| | | VV+VH | 95.0 | 100.0 |
| | | VV | 84.4 | 99.3 |
| 2019-09-06 | 2019-09-06 | VH | 86.2 | 100.0 |
| | | VV+VH | 88.0 | 100.0 |

may be decreased to the point that no distinction is possible also by visual means. Such a case can be observed in Fig. 7a),
where hardly any open water surfaces could be detected by the algorithm. The scene was acquired on a windy day during
which average wind speed measured at nearby Jamestown Regional Airport exceeded the 99% quantile of the recorded daily
averages since 1990 (source: GSOD). Especially larger water bodies are not visible in the co-polarised backscatter image due
to the low contrast. The water body delineation based on VH-polarised data, however, did not display the same issues. While
the backscatter from larger open water bodies still shows some influence of water surface roughening, this fact inhibited water
extent delineation to a far lower degree (Fig. 7b).

The effect of wind on SAR backscatter from water surfaces was a frequently encountered problem in this study, especially
for VV-polarised imagery. VH data were shown to be less affected by this problem, which is in line with our expectations and
findings available in literature (Henry et al., 2006). However, as the cross-polarised signal is dominated by volume scattering
(Richards, 2009), areas with sparse vegetation often resemble water surfaces in the resulting imagery (Twele et al., 2016).
Hence, its full potential lies in the complementary use together with co-polarised data as it was demonstrated here.

Overall, the proposed approach for water extent classification represents an efficient way of fusing SAR backscatter with
topographical information via the derivation of HAND. Such probabilistic approaches have drawn some attention in recent
years to combine SAR imagery with multi-temporal and ancillary sources of information (D'Addabbo et al., 2016; Li et al.,
2019). D'Addabbo et al. (2016) used interferometric coherence, a geomorphic flood index and Euclidean distance from a river
to incorporate a-priori information on the probability of an area becoming flooded during an inundation event. The authors note
that, when using the ancillary information, both false and missed alarms were reduced with respect to the use of SAR intensity
data only.

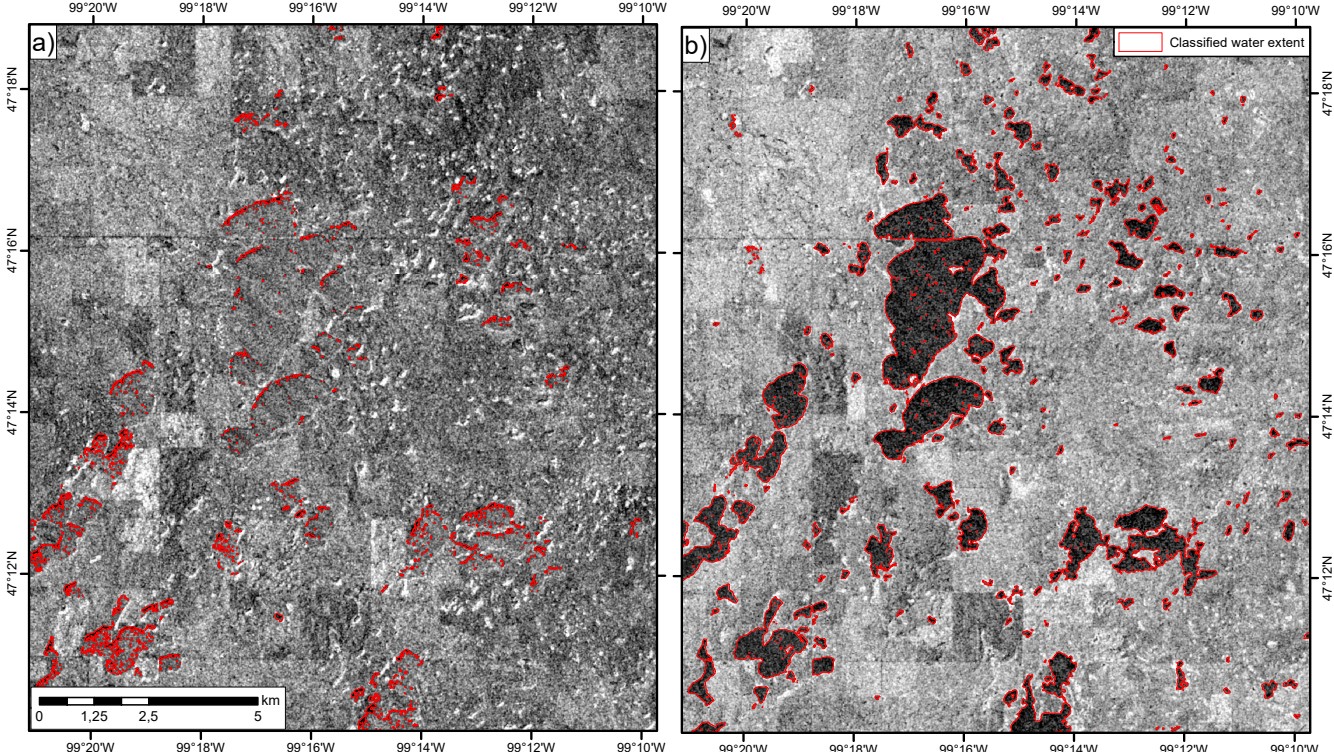

**Figure 7.** Backscatter in a) VV and b) VH polarisation and derived water bodies on 12 October 2019.

## 3.2 Surface water dynamics

Throughout the study period the number of potholes with open water surfaces strongly varied between ca. 2,300 and more than

5,000. In general, the number of water bodies, the total water area and the median area per water body were relatively stable between 2015 and 2019, whereas 2020 differed considerably from the rest of the observation period (Fig. 8a). A number of outliers occur in the time series. For example, the image acquired on 12 October 2019, when heavy winds influenced the water retrieval (Fig. 7) has a lower number of water bodies and total water area and higher average water area per detected water body indicating that only larger objects could be detected. In addition, e.g., in October of the years 2015, 2017, 2018 and 2020,

the number of water bodies and their median area showed sudden changes due to wind on the water surface or emerging ice cover which limited the algorithm's capability to accurately delineate the surface water extent. As a result, large water bodies were only partially classified as water and, therefore, higher numbers of small water bodies were erroneously identified. In the following discussion, the results for these dates will not be taken into account.

Between 2015 and 2019, the number of water bodies displayed a seasonal behaviour with highest numbers in spring, then

reaching an annual minimum in late summer and increasing again during September and October (Fig. 8a). During the same period, the total water area varied between ca. 14,000 ha and 16,000 ha. In the first half of the study period, total water area





declined from ca. 16,000 ha in 2015 to 15,000 ha in 2017. Seasonal water dynamics were also pronounced. In most years, total water area declined from spring throughout summer until autumn. This is especially noticeable in 2017, when water area declined from ca. 15,400 ha in May to ca. 13,180 ha in October. The subsequent two wetter years again differ from each other

in terms of intra-annual dynamics of water area. In 2018, total water area remained relatively stable throughout the snow-free period while 2019 started with higher values which then declined until mid-September by ca. 1,500 ha. This was followed by a steep increase until water area reached its maximum of the study period up to that point at ca. 16,500 ha (Fig. 8b). This behaviour coincides with the exceptional spring floods of that year and the wet October leading to widespread flooding along the James River and other regions of ND (Umphlett, 2019). The median area per water body (Fig. 8c) was similar to the inverse

of the number of water bodies. Potholes tended to have larger water areas in summer than in spring and autumn. In combination with the seasonal decrease in water area and number of water bodies this suggests that smaller water bodies dry out during summer. In addition to the strong increase in total water area at the end of 2019, which is not found in other years, we can also observe a decrease in median water area per pothole and a larger number of water bodies, suggesting that a large number of small potholes filled due to the intense storm event. The year 2020 differs from the rest of the study period in all three

indicators. The number of water bodies and the total water area are consistently higher than in all previous years, with both indicators surpassing the maximum values of those years during all the observed months in 2020. At the same time, the median area per water body was smaller in 2020 than in previous years, indicating a much larger abundance of small water bodies. The decrease of both total water area and the median area along with the stable or even increasing number of water bodies suggests that small water bodies remained present throughout 2020.

The persistence of small water bodies throughout 2020 becomes obvious when looking at the contributions of water bodies of different sizes to the total water area. Fig. 9 shows the total water area of water bodies from four different size classes. In the following, water bodies > 1 ha will be referred to as large water bodies, water bodies < 1 ha as small water bodies. Large water bodies accounted for most of the total water area in the Pipestem Basin. The water area in these size classes (1 to 8 ha, > 8 ha), with the exception of the aforementioned dates affected by wind and ice cover, was rather stable until 2018 and showed little

seasonal variability (Fig. 9a and b). The water area accounted for by small water bodies, however, showed a clear seasonal pattern with a decrease in water area from spring throughout summer (Fig. 9c and d). In 2019, water area was higher in spring than in summer also in the two larger size classes. During the storm event of October 2019, water area increased in all four size classes. Large water bodies accounted for 88.7% of the total increase in water area of almost 2,000 ha between 18 September and 24 October 2019. During the wet year 2020, water area of large water bodies showed an adverse behaviour from small

water bodies. While the water area in the larger two size classes declined between May and July, the total area of small water bodies stayed relatively stable and increased in autumn.

The strong increase in water area and the emergence of small wetlands in October 2019 with respect to an earlier year can also be seen in the water extent maps in Fig. 10. In 2016 (Fig. 10a), a decrease in the extent of water bodies is visible from early summer (green colour) towards August (dark blue and yellow). The decreased water extent leads to a disconnection of some

water bodies into several smaller ones. Between August and October, water extent only increases by a small amount, which can be seen in red. During 2019, in contrast, the open water extent visible in June (green colour) is small while in later months



**Figure 8.** a) Number of water bodies, b) total water area and c) median water body size across the Pipestem Creek basin for each of the Sentinel-1 scenes. Lines show LOESS smoothing functions along with 95% confidence intervals.

August and October (orange and red colour, respectively) water extent increases and, especially in October, many wetlands are beginning to merge (Fig. 10b). This disconnecting and merging behaviour illustrates that an increase in overall water area not necessarily translates into an increase in the number of small water bodies and vice-versa. This may explain some of the

seemingly non-corresponding temporal patterns seen in total water area, number of water bodies and median water body size in Fig. 8.

Using the time series of water extent maps derived from Sentinel-1 imagery it was possible to track changes in total water area, number of water bodies and median water body area. In general, the temporal patterns found in these variables followed the expected seasonality. In a modelling study, Liu and Schwartz (2011) Liu and Schwartz [9] reported intra-annual dynamics

with the number of small water bodies declining from spring throughout summer, whereas the number of larger water bodies







**Figure 9.** Total water area of water bodies with extents a) > 8 ha, b) 1 to 8 ha, c) 0.2 to 1 ha, d) 0.05 to 0.2 ha. Lines show LOESS smoothing functions along with 95% confidence intervals.

remained stable throughout the year. Our results covering a period of six years corroborate these findings. In most years, the contribution of small water bodies to total water extent declined from May towards the end of summer and then increased again until the end of October. The area of larger water bodies, however, remained stable. Along with the typically declining number of water bodies and the increase in median water body size, this suggests that a large number of small water bodies falls dry

during the summer months when evaporation rates are high. In the following year, these small potholes typically re-fill after spring rains and snowmelt.



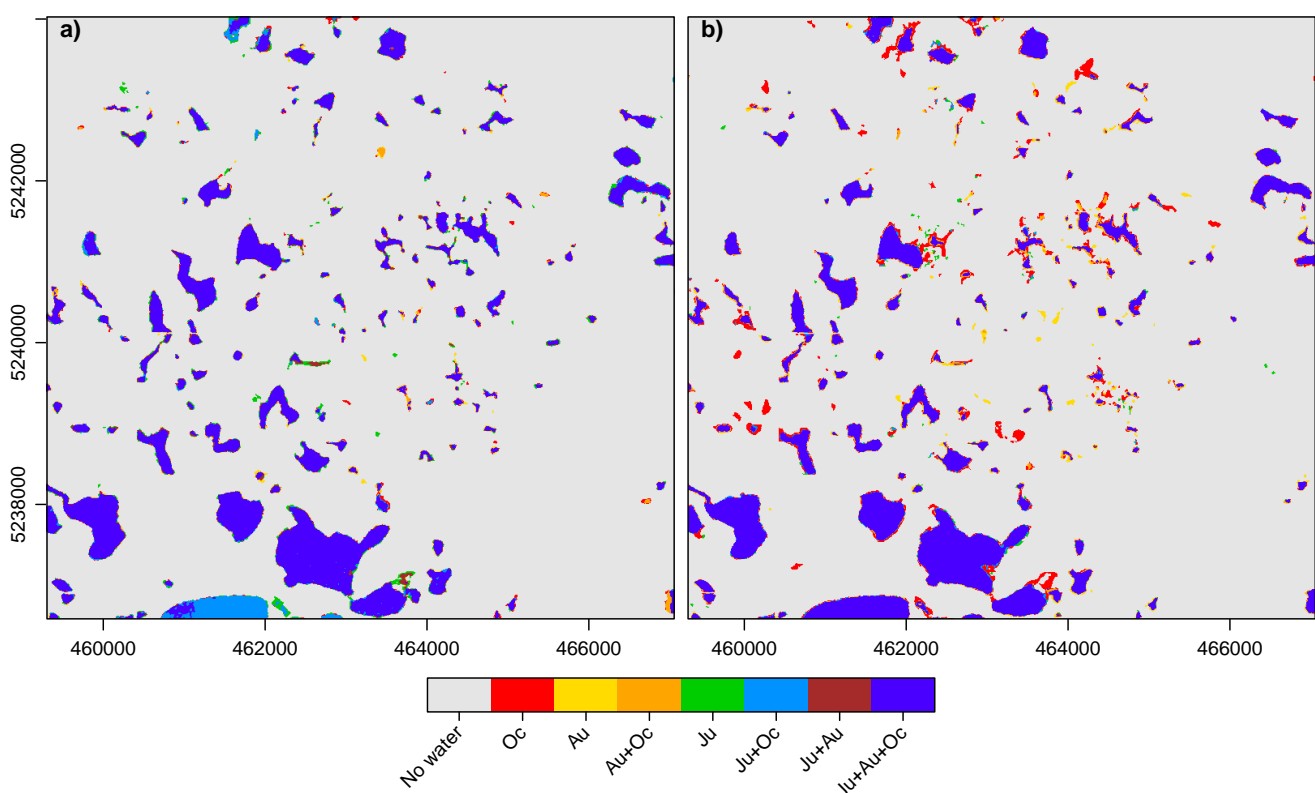

**Figure 10.** Water extent dynamics in a) 2016 and b) 2019 in a subset of the study area. Observation times refer to 5 July 2016 and 14 June 2019 (Ju), 10 August 2016 and 13 August 2019 (Au), 27 October 2016 and 24 October 2019 (Oc), respectively. + symbol denotes water detected at multiple observation times. Dark blue areas were detected as water at all three dates. Scales are in UTM (zone 14) coordinates.

To our knowledge, this study is the first time that surface water dynamics in the PPR have been monitored using Sentinel-1 over a longer period covering both dry and wet years. This enables us to also support findings from previous studies (e.g. Liu and Schwartz, 2011) on inter-annual differences in wetland extent. While intra-annual changes in water extent have been found mainly for small water bodies, larger wetlands should only change on an inter-annual time scale. Our results demonstrate

that the contribution of wetlands both larger and smaller than 1 ha to the total water extent is in line with what would be expected from meteorological indices indicating water availability, such as PDSI (Fig. 2). During the extremely wet period in late 2019/early 2020, large water bodies showed significantly higher water extent values than during the rest of the time period. It is noteworthy that these increased water extents in larger wetlands decreased again relatively quickly along with the

decrease in PDSI during 2020, whereas the extent of small water bodies remained relatively stable. This may suggest that larger wetlands can act as a water storage during wet periods but then return to their formerly stable extent by releasing water to the



drainage network. Small potholes, however, tend to be more geographically isolated, i.e., do not have well-defined inlets or outlets and their water balance is mainly controlled by vertical fluxes, such as rainfall, evaporation and drainage to the sub-soil (Cohen et al., 2016). They may persist as long as meteorological conditions are wet enough to support them. During most of 2020, PDSI was above 2 in the study region, which indicates such conditions. Furthermore, the snowpack during the winter of 2019/2020 was thicker than normal (Umphlett, 2019) and the water released after melting fed many of the smaller potholes leading also to a higher number of smaller water bodies.

The time series of water extent and number of water bodies also reveal differences between the 2019 flood events reported in the literature (Umphlett, 2019). At the time of writing, we could not find an analysis of the effects of these extreme events on wetland extents in the literature. While extensive floods have been reported for spring 2019 across the Midwest (Umphlett, 2019; NOAA, 2019) (Umphlett, 2019; NOAA, 2019), which are visible as peaks in the discharge time series, the autumn event led to much higher flood peaks in the Pipestem Creek (Fig. 2). During the spring event, the extent of large water bodies showed a lower increase with respect to the summer months than during the autumn event. The extent of small water bodies also increased during the second event, however, not as much as in spring. The results reported here show merging of smaller into larger wetlands, which may contribute to this behaviour. This finding, which is somewhat in contrast to our expectation that small potholes replenish more quickly than larger ones, may also suggest that larger water bodies contributed more to floodwater runoff than small water bodies. Inter-annual changes in prairie wetland extent have been tracked using Landsat data (Vanderhoof et al., 2016; Krapu et al., 2018; Rover et al., 2011) and high-resolution aerial imagery (Wu and Lane, 2017; Wu et al., 2019), however, the limitations due to cloud cover and the long intervals between acquisition of NAIP data typically do not allow to reproduce extreme events. The analysis of SAR time series unaffected by cloud cover and with high temporal resolution may help to understand the complex threshold behaviour which characterises catchments in the PPR (Shaw et al., 2013).

A major limitation encountered in this study for the monitoring of wetland dynamics during extreme events is the rather low temporal resolution of Sentinel-1 over the study area. Imagery over the study area currently is only acquired by one of the two satellites of the Sentinel-1 constellation and only along ascending passes. This limits the acquisition interval of the entire catchment to the revisit time of 12 days. Additionally, imagery was not acquired during every overpass, further reducing temporal resolution. Obviously, this time span is too large to resolve flood events caused by storm or rain-on-snow events. The combination of Sentinel-1 data with other SAR sensors, such the Radarsat Constellation Mission or the future NASA-ISRO SAR (Kumar et al., 2016) mission, may help to mitigate this problem in the future. In the present study, only water extent, which is directly observable by satellite imaging systems was analysed. However, for many applications, such as water availability assessment or flood management, also surface water storage would be of interest. Pothole bathymetry data only exist at very limited scales (e.g. Mushet et al., 2017). While empirical relationships between water surface and stored volume exist for prairie potholes of different size classes (Gleason et al., 2007) validating such estimates is difficult due to missing reference data. The planned Surface Water Ocean Topography (SWOT) mission (Biancamaria et al., 2016) may help to provide such information in the future.



## 4 Conclusions

In this study, a novel approach for retrieving dynamic open water extent in prairie pothole wetlands from dual-pol Sentinel-1 SAR data was presented. Using a Bayesian framework, topographic information was integrated in the retrieval processed via HAND. The results demonstrate that the approach was successful in mapping changes in water extent in prairie potholes when
their location was known *a priori*. The inclusion of topographic information, at least in some cases, helped to mitigate the adverse effects of non-water areas resembling water surfaces due to low backscatter and of wind roughening the water surface. The impact of the latter factor was further decreased by the combination of co-polarised and cross-polarised SAR data as the latter are typically less affected by surface roughness. The obtained time series of total water area, number of open water bodies and median wetland area covering a time period of six years showed clear intra-annual as well as inter-annual patterns. The
different responses of small (< 1 ha) and large (> 1 ha) wetlands to an extremely wet period lasting from 2019 to 2020 were hypothesised to be at least partly the result of the different degree of connectivity between small and large potholes within the catchment. Notwithstanding the value of the retrieved dynamic wetland maps, limitations persist with respect to the effect of wind on SAR backscatter from open water and the rather long revisit cycle of Sentinel-1 of 12 days over large parts of the PPR. However, the value of Sentinel-1 for this application will further increase with the time period covered by this long-term Earth
observation programme.

*Author contributions.* Stefan Schlaffer designed the study and developed the Sentinel-1 classification algorithm, processed the data, analysed the results. Marco Chini extensively contributed to the development of the classification algorithm, the structure of the manuscript and the discussion of the results. Wouter Dorigo extensively contributed to the structure of the manuscript and the discussion. Simon Plank contributed to the data processing and to the discussion of the results. Stefan Schlaffer prepared the manuscript with contributions from all
co-authors.

*Competing interests.* The authors declare that they have no conflict of interest.

*Acknowledgements.* The authors would like to thank the organisations providing the data used in this study: Copernicus (Sentinel-1 data), the USGS (runoff data), NOAA NCDC (precipitation data), the National Agriculture Imagery Program administered by the USDA Farm Service Agency (aerial imagery), the North Dakota State Water Commission (topographic data) and Copernicus Programme (Sentinel-1 data), the
USDA NASS (Cropland Data Layer).



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





# Appendix A: Validation dataset

**Figure A1.** NAIP false-colour composites acquired in a) 2016, b) 2017 and c) 2019. Green dots show sample points used for validation of water bodies derived from Sentinel-1. d) Location of the NAIP footprints inside the study area.