# Peer review of "Monitoring Surface Water Dynamics in the Prairie Pothole Region of North Dakota Using Dual-Polarised Sentinel-1 SAR Time Series"

_Hydrology and Earth System Sciences, 2021_

## Author Comment (AC2)

We wish to thank the reviewer for the helpful comments. You can find our replies below in red.

The paper describes an experiment to monitor numerous small lakes in the PPR region in USA, which expand/shrink and increase/decrease in number according to precipitations.
The authors used time series of the Copernicus Sentinel-1 SAR mission, together with a high-resolution DTM, to detect the extension of water on each of the time series images, which in this area reach a temporal sampling rate of 12 days.

The paper appears well written and describes a great deal of work. The choice of data and processing is effective, and the conclusions appear convincing and interesting. I particularly appreciate the use of the VH polarization channel as an additional information source, as it appears somewhat underrated in the current literature.

The processing algorithm is based on a Bayesian framework to derive maps of posterior probabilities that each pixel is inundated in each time slice, relying on priors and, in this case, external ancillary data (the DTM) for the a priori water and non-water distributions. The Bayesian framework allows to derive such maps as real numbers (the actual probability value), while the authors in the end perform some thresholding to arrive at binary water/non water maps. Although clearly this representation is more straightforward to interpret by a wider audience, I find that a good deal of information is actually lost with the thresholding. Dealing with a continuous indicator such as the final water probability allows to retain some form of confidence measure about the presence of water given the other sources of information, which is somewhat more complete than a binary map. Moreover, it allows using a different set of evaluation tools. For instance, ROC analysis allows to derive detection efficiencies (e.g. the AUC), as well as threshold values to binarize the maps, optimized with respect to the different pixel populations. I would like to know the authors' opinion on this issue, and at least see some comments in the manuscript justifying their choices.

This is a very helpful comment. It is true that information is lost when binarizing the probability maps. In the end we chose to binarize them in order to make it easier to compute the water areas reported in section 3.2 as this was our main interest in this paper. The information on water probability is however useful.

The two maps below show the posterior probabilities, $p(w|\sigma^0_{VH})$ and $p(w|\sigma^0_{VV})$, for VH (in green) and VV (in blue) polarisation, respectively, for the binary maps shown in Figure 6c,d of the manuscript. As it can be seen, the borders between high and low water probability are quite crisp in most of the cases. This is due to the inclusion of the HAND-derived p(W) prior probability. There are some areas with intermediate p values, mainly between high-probability wetlands.

This is obviously important for illustrating our conclusion that the integration of HAND helps with the avoidance of false positives. We can add these maps to either Fig. 6 or to the Appendix. In the text, we will add the aforementioned considerations in the paragraph in lines 297-310. In the discussion around lines 346-352 we can also add that, in this case, we chose fixed thresholds for binarising the probabilistic maps to simplify the process and mention the advantages of ROC analysis to derive thresholds.

[Figure]

Apart from this "methodological" point, I have only some minor comments as follows.

Line 226: it would be useful to clarify what are these two "histogram portions"... maybe add an equation (as for eqs. 5 and 6)...?
The term "histogram portions" is not well chosen. We propose to change this sentence to "where $\mu_1$ is the mean of all $\sigma^0_p \leq \tau_i$ and $\mu_2$ the mean of all $\sigma^0_p > \tau_i$. $\sigma_1^2$, $\sigma_2^2$ are their variances." $\tau_i$ is the initial Otsu threshold and will be defined in line 223.

Line 236: "where... respectively". This sentence is not very clear to me.
The word "respectively" is used erroneously here, we will remove it. In addition, we will change line 231 to "The parameters of the backscatter distributions for water, w, and land, l, for each pothole were estimated as …"

Line 246: this mention of Bayes' theorem could be moved somewhere before eq. (4) for clarity.
We propose to mention it in line 114 to have it in the introduction so that the reader knows what to expect: "We use a probabilistic approach based on Bayes' theorem combining SAR backscatter…"

More generally, the procedure described in sect. 2.3 is a little bit involved, so it would be beneficial to add a flow-chart or a pseudo-code algorithm description to better clarify how it works.
Figure 3 shows a flowchart of the overall procedure but it is not well bound into the text. We propose to add some details on the "pothole specific threshold retrieval" and to number the key elements of the flowchart. Like this the different elements can be easily referred to in the text and it will be clearer for the reader which part of the overall processing is being described.

Figure 6: labels c) and d) are not visible (black background). More generally, labels should be probably enlarged a little for visibility. How are the false color composites in panels e) and f) obtained exactly?
Thank you for pointing this out. We will make the labels white and enlarge labels in general. We will modify the caption: "false-colour composites (red – near infrared, green – red, blue – green) of NAIP imagery acquired 4 July 2016."

Line 394: "adverse" -> "opposite"?
We will change it.

Figure 8: please add some explanation and possibly a reference for the LOESS acronym and meaning.
LOESS stands for "locally estimated scatterplot smoothing" and is described in Cleveland et al. (1992). It is also often referred to as the Savitzky-Golay filter. We will add some explanation in the text (section 2.4).

Line 409: correct citation.
We will correct it.

Lines 407-416: I believe this period contains points which are partially repeated later in the following paragraph. You may want to consider merging this with the subsequent text.
In response to the comments of another reviewer we propose to reorganise section 3.2. By having a clearer structure it will become easier to avoid repetitions such as these.

We propose to split the section into two subsections: in 3.2.1, we will describe the results by explaining the temporal dynamics of number of water bodies, total water area and median water extent as observed using Sentinel-1 data (Figures 8-9). In 3.2.2, we will discuss our observations in the context of hydrometeorological conditions (i.e. discharge, precipitation) and discuss their impacts on the observed intra-annual surface water extent dynamics. After this, we discuss the observed inter-annual dynamics in the light of dry and wet periods as indicated by PDSI. This structure should be much clearer.

**References**:

W. S. Cleveland, E. Grosse and W. M. Shyu (1992) Local regression models. Chapter 8 of Statistical Models in S eds J.M. Chambers and T.J. Hastie, Wadsworth & Brooks/Cole.

---

## Author Response (AR1)

**Reviewer #1**

We wish to thank the reviewer for the helpful comments. Please find our replies below in red.

The authors mapped the open water body in the PPR using SAR images and high-resolution DTM data. Time series of water body maps were obtained and the dynamics of open water bodies were analyzed. Generally, this manuscript is well prepared with good structures and figures. However, I have some major concerns about the current work, as:

1) The validation (or accuracy evaluation) of the method. Currently, the authors evaluate the accuracy from a viewpoint of classification, using user accuracy and produce accuracy for the pixels selected. However, as the authors analyzed in section 3.2, what the hydrologist cares about is the number and area of the water body. So, the authors are suggested to validate (evaluate) their method by comparing the area and number estimated from SAR with those from NAIP.

We agree that this would be interesting information. However, a classification of NAIP mosaics, such as the ones shown in Fig. A1, using either a supervised or unsupervised approach into water bodies and non-water areas is also prone to errors and would require an accuracy assessment of its own. Manual delineation would likely not find all the different water bodies in the mosaic. Also, it is not clear which water bodies should be counted. Prairie wetlands can be as small as a few square metres, i.e. they may be visible in the NAIP imagery but not in Sentinel-1 images.

Hence, we opted for a "traditional" point sample-based accuracy assessment for the manuscript. In response to your comment, we classified one of the NAIP mosaics using a supervised approach. You can find the methods and results below illustrating an underestimation of the number and total area by Sentinel-1. However, it also shows the challenges in connection with creating such a reference dataset from optical airborne data, e.g., false positives in areas with low vegetation cover or tree shadows. Such a small analysis cannot properly address these issues and would warrant a study of its own. We therefore think that this validation is beyond the scope of our study.

Classification of NAIP mosaic and comparison with Sentinel-1 water bodies:

We classified the 2019 NAIP mosaic (shown in Fig. A1c) using a Random Forest (RF) classifier that was trained using 350 of the 400 reference points used in the original accuracy assessment. The remaining 50 points were set aside for model testing. In addition to the four NAIP bands, also NDVI was used as a feature. The classified raster contained many false positives in areas with little vegetation, such as recently ploughed fields and tarmac, as well as areas shadowed by tall vegetation and buildings (Figure R1). For this reason, we intersected the RF-derived areas with high water probability ($p > 0.8$) with the original, LiDAR-derived, potholes and retained those objects as seeds. These seeds were used for region growing with a relaxed probability threshold ($p > 0.5$). For comparison with the coarser-scale Sentinel-1 product, the NAIP product was upsampled to 10 m, whereby only pixels with at least 50% water coverage were retained.

The RF model overall accuracy was estimated at 96% using the test dataset of 50 points. The classification based on NAIP yielded a number of 449 water bodies covering an area of 1967 ha. In contrast, the Sentinel-1 wetland product over the same sub-region contained 243 water bodies with a total area of 1634 ha. However, in the size class > 1 ha, Sentinel-1 had 183 water bodies, while the NAIP-based result contained 185. This demonstrates that especially smaller water-filled potholes remained undetected in the Sentinel-1 product. This seems plausible due to the higher proportion of

mixed pixels along the edges as well as shallower water depths which may lead to vegetation protruding through the water surface. The large amount of small erroneous water pixels in Figure R1 suggests, however, that at least some of the NAIP-based small water bodies are false positives, even after the post-processing steps described above.

[Figure]

*Figure R1: Subset of random forest water probabilities > 0.5 for the NAIP mosaic of 2019.*

2)L332-345, the authors explained the different performances of different polarization combinations. However, this work can be done more physically by introducing the Radar functions. Actually, the authors can start from the radar functions and then build their algorithm on the basis of microwave radiative transfer theory.

Parameters can be retrieved from Earth Observation (EO) data using inversion of radiative transfer models or, as in this case, using statistical approaches. In this case, the use of radiative transfer (RT) modelling would have been very difficult to realise given the large diversity of the background (i.e., non-water) classes along with a high temporal variability of backscatter within each class (after all, we analyse six years of data). This includes different vegetation types and stages, different crop types on agricultural areas, as well as vastly different soil moisture conditions, which all influence backscatter. Furthermore, as mentioned in the manuscript, wind is an important factor on the temporal variability within the water class itself. The parameterisation of RT models would have been extremely challenging and would likely not have yielded robust results as were obtained here. Moreover, a large amount of ancillary data would have been required (e.g., accurate land cover, crop type, vegetation growth stage, soil moisture status, roughness) which are either not available or have their own, often considerable, uncertainties. For these reasons, we did not make changes to the manuscript regarding this comment.

3) the abstract is too long, please shorten it.

We agree. We shortened the abstract from ca. 480 to 380 words:

"The North American Prairie Pothole Region (PPR) represents a large system of wetlands with great importance for biodiversity, water storage and flood management. Knowledge of seasonal and inter-annual surface water dynamics in the PPR is important for understanding the functionality of these wetland ecosystems and the changing degree of hydrologic connectivity between them. Optical

sensors widely used for retrieving such information are often limited by their temporal resolution and cloud cover, especially in the case of flood events. Synthetic aperture radar (SAR) sensors can potentially overcome such limitations. However, water extent retrieval from SAR data is often impacted by environmental factors, such as wind on water surfaces. Hence, for reliably monitoring water extent over longer time periods robust retrieval methods are required.

The aim of this study was to develop a robust approach for classifying open water extent in the PPR and to analyse the obtained time series covering the entire available Sentinel-1 observation period from 2015 to 2020 in the hydrometeorological context. Open water in prairie potholes was classified by fusing dual-polarised Sentinel-1 data and high-resolution topographical information using a Bayesian framework. The approach was tested for a study area in North Dakota. The resulting surface water maps were validated using high-resolution airborne optical imagery. For the observation period, the total water area, the number of water bodies and the median area per water body were computed. The validation of the retrieved water maps yielded producer's accuracies between 84 % and 95 % for calm days and between 74 % and 88 % on windy days. User's accuracies were above 98 % in all cases, indicating a very low occurrence of false positives due to the constraints introduced by topographical information.

The observed dynamics of total water area displayed both intra-annual and inter-annual patterns. In addition to differences in seasonality between small (< 1 ha) and large (> 1 ha) water bodies due to the effect of evaporation during summer, these size classes also responded differently to an extremely wet period from 2019 to 2020 in terms of increase in number and total area covered. The results demonstrate the potential of Sentinel-1 data for high-resolution monitoring of prairie wetlands. Limitations exist related to wind inhibiting correct water extent retrieval and due to the rather long acquisition interval of 12 days over the PPR."

4) some figures, like figure 5, are suggested to add scales and the compass. Since not all readers are familiar with UTM(zone 14)

We added a scale bar and a north arrow. We also spelled out UTM in the caption (requested by another reviewer).

5) The authors are suggested to pay attention to the usage of abbreviations, for example, please define GRASS and dual-pol.

Thank you for pointing this out. Since we only use the term "dual-pol" twice, we changed it to "dual-polarised". GRASS GIS is a widely used GIS package and commonly referred to as GRASS GIS whereas the full name "Geographic Resources Analysis Support System" is much less known. The full name is given in the Reference with a link with further information (Line 534).

**Reviewer #2**

This article is novel in its approach to coping with the difficult problem of separating surface water from (possibly wet) vegetated land in a small catchment of 2770 sq.km in the Prairie Pothole Region in North America. The data to make the scheme feasible was obtained from the orbiting satellite Sentinal-1A, at a resolution of about 20m, sampled during the summer and autumn months, so were not blanketed by snow. The authors developed the tools to obtain valid images to work from.

The text is very well laid out and informative – there are only a few places that need repair and I am glad to say that I read every word so that [perhaps not to the authors' taste] I made some very minor alterations, where I deemed necessary. The math is well set out, but I would like the equations inset from the margins; however that's a minor issue of layout. The Figures are good and easy to understand. Regretfully some of them are placed up to 3 pages away from their first mention, which is irritating. There is however the odd glitch, like Fig. 9, where the caption and label are in in hectares, and the image legends are in sq m!! Fortunately, I could not find any others.

I judge that this paper is worth publishing after repair, so recommend minor corrections. I am returning the version of the paper that I have marked up, attached to this review. In addition, I will import the more substantial of my remarks to be listed below my signature, which is my wont.

Geoff Pegram

18 August 2021

Dear Geoff,

We would like to thank you for your helpful comments. We absolutely agree with the observation that some of the figures and Table 1 are placed too far away from their first mention. This may also be a result of the fact that we may not have fully exploited the Latex options for placing the figures and so much of the placement was actually done automatically. We changed this in the revised version (final version will anyhow have different 2-column layout).

We are responding to your individual comments below in red.

++++++++++++++++++++++++++++++++++++++++++++++++++++++++++++++++

Comments, line-by-line from the article, followed by # and my suggestions

Abstract. The North American Prairie Pothole Region (PPR) represents a large system of wetlands with great importance for biodiversity, water storage and flood management. Knowledge of seasonal and inter-annual surface water dynamics in the PPR is important for understanding the functionality of these wetland ecosystems and the changing degree of hydrologic connectivity between them.

**Try and make this more interesting so that, although it is informative as an introduction, please make your abstract (or intro) more descriptive - Wiki's description is a good springboard – your paper is not overlong:**

"The Prairie Pothole Region (PPR) is an expansive area of the northern Great Plains that contains thousands of shallow wetlands known as potholes. These potholes are the result of glacier activity in the Wisconsin glaciation, which ended about 10,000 years ago. The decaying ice sheet left behind depressions formed by the uneven deposition of till in ground moraines. These depressions are called potholes, glacial potholes, kettles, or kettle lakes. They fill with water in the spring, creating wetlands, which range in duration from temporary to semi-permanent. The region covers an area of about 800,000 sq. km and expands across three Canadian provinces (Saskatchewan, Manitoba, and Alberta) and five U.S. states (Minnesota, Iowa, North and South Dakota, and Montana). The hydrology of the wetlands is variable, which results in long term productivity and biodiversity. The PPR is a prime spot during breeding and nesting season for millions of migrating waterfowl. [Wikipedia]"

Two other reviewers suggested substantial shortening of the abstract but we are happy to use your ideas for the introduction. We changed Line 36 to 40 to the following (changes highlighted in yellow):

"The Prairie Pothole Region (PPR) of North America covers an area of over 780,000 km² in the Great Plains of the Northern USA and Southern Canada. The region is characterised by millions of shallow depressions formed during glacier retreat at the end of the last glacial period, when glacial till was unevenly deposited in ground moraines. These depressions contain  wetlands whose areas vary between one square metre and several square kilometres. They can store considerable amounts of water during rainfall events, which contributes to flood mitigation in downstream populated areas (Huang et al., 2011b). The wetlands of the region are of great importance for the waterfowl population of North America (Mitsch and Gosselink, 2000)."

In turn, we removed a repetition of the above in Line 129.

27….. due to the rather low temporal resolution of 12 days over the PPR.

**You make this point later on in the text (lines 97, 155, 451 & 473) but the reader is left unsure as to whether this data is sampled 12 days apart, or averaged over that interval**

For this part of the PPR, the Sentinel-1 observation scenario foresees only acquisitions using the second satellite of the pair (Sentinel-1B), hence new imagery is acquired only every 12 days, not every 6 days as would be the case if both satellites were acquiring data. We changed "low temporal resolution" to "long acquisition interval" to clarify this.

90 attention, as co and cross-polarised data

**Unsure what 'co' means. Aha! found it on the web - does it make sense? :- "Co-polarization is the antenna's radiation in your desired directions. Whereas cross-polarization is the antenna's radiation in the unwanted directions, i.e the cross-polar is basically considered as a dissipation in antenna radiation."**

We changed this sentence as follows to explain these terms: "In particular, the analysis of polarimetric SAR data has received attention as data acquired using different polarisations for sending and receiving (cross-polarised data) respond differently to scattering mechanisms, such as surface and volume scattering, than co-polarised data (same polarisation used for sending and receiving)."

184: composites of the images are shown in Appendix A (Fig. A1).

**Fig. A1 is lonely in an appendix, but informative - I recommend your replanting it about here; there's enough room for it in this medium sized paper**

Our idea behind placing this figure in the appendix was due to a) the paper already has 10 figures in the main text, and b) in our opinion, the exact distribution of the sample dots over the images does not add much information that the reader needs to understand the approach. As an alternative to placing the entire figure in the main text, we decided to show the footprints of the aerial images in Figure 2 and add a reference to it in section 2.2.4. In response to the comment of another reviewer, we added another figure to the Appendix, so there are now two figures there.

225: …… $D = p2|\mu1 - \mu2| \_2\,1 + \_2\,2$ ,……… (1)

**For easier readability, please indent your 7 equations at both ends**

We agree but the text was set using the HESS Latex template which we followed so I don't think we have much choice here. I checked some other preprints and found the same style without indentation there.

313 water extent (Table 1).

**Too far ahead on page 16**

We changed the placement so that it appears now two pages ahead on page 15. Page 14 is already taken up by a Figure.

Figure 5. Map of predicted p(W). Scales are in UTM (zone 14) coordinates.

**Please tell us a bit more about this interesting figure in the caption which should be expanded. The caption should redefine the acronyms and symbols (p(W) and UTM).**

This suggestion applies to all figure captions. It helps the first quick scan for the tentative reader.  I had to make a list of acronyms before I started reading through critically, as I can't retain them in all my head!

We changed the caption as follows: "Map of prior water probabilities p(W) predicted from Eq. (7). Scales show coordinates in the Universal Transverse Mercator system (UTM zone 14)."

329: … HAND….

**How do you measure HAND, and to what precision?  I see that you mention it in subsection 2.2.2 Topographical data, but you don't elaborate.**

HAND is determined by first computing the drainage direction for each DTM raster cell. For each cell, the algorithm follows the drainage direction raster until a cell belonging to the drainage network is reached. In this case we used both the drainage raster output by r.watershed (GRASS GIS; https://grass.osgeo.org/grass78/manuals/r.watershed.html) and the potholes as drainage pixels. Then the height difference between the original cell and the associated nearest drainage pixel is

taken as HAND value and assigned to the original cell from which the algorithm has started. Further details are given by Rennó et al. (2018). We added a more extensive explanation to 2.2.2. (Line 169-170): "HAND is defined as the difference in elevation between a given DTM cell and the nearest cell pertaining to the drainage network (Rennó et al., 2018). For this purpose, the flow direction was determined using the D8 method. The algorithm then follows the flow direction raster until reaching a cell pertaining to the drainage network and computes the height difference between the drainage cell and the original starting cell."

Figure 7. Backscatter in a) VV and b) VH polarisation and derived water bodies on 12 October 2019

**That's pretty smart**

365 Fig.8a

**Please rearrange Figs 8 & 9 closer to first mention - they are up to 2 pages distant - I have to split the document to follow text.  Not good if it's a printed copy ...**

We changed the placement of the figs. so that they appear on the same page or the page following the first mention.

398: Fig. 10

**3 pages ahead ... and I have to expand the figure to 300% to find the tiny yellow patches; can't you take a small clip and park it on the empty space like I've done?**

We will changed the placement, it is now on p. 21, the first mention on p. 20. We provided also a zoom into a subset with patches. We darkened the background colour to provide more contrast to the yellow patches.

Fig. 9

**The caption and label are in hectares, the image legends are in sq m!! That's confusing - please fix**

The y axis labels show hectares. The figure shows the sum of the areas of all water bodies belonging to each size class. This is what we mean by "Total water area" in the caption. We changed it to "Summed areas of water bodies in size classes a) > 8 ha …".

447-460

**That last paragraph is a good summary**

475 … programme

**there is a choice here – what about 'program'?**

'In American English, program is the correct spelling. In Australian and Canadian English, program is the more common spelling. In British English, programme is the preferred spelling, although program is often used in computing contexts.' [Grammarly]

We went with the spelling that the "Copernicus Programme" uses, of which Sentinel-1 is a part. They refer to themselves as programme (e.g. 1st sentence here: https://www.copernicus.eu/en/about-copernicus).

476  Author contributions

**I like this list of the contributions of the team.  Nice job.**

++++++++++++++++++++++++++++++++++++++++++++++++++++++++++++

https://hess.copernicus.org/preprints/hess-2021-330#RC2

References:

Rennó, C. D., Nobre, A. D., Cuartas, L. A., Soares, J. V., Hodnett, M. G., Tomasella, J., and Waterloo, M. J.: HAND, a new terrain descriptor using SRTM-DEM: Mapping terra-firme rainforest environments in Amazonia, Remote Sensing of Environment, 112, 3469–3481, 2008.

**Reviewer #3**

We wish to thank the reviewer for the helpful comments. You can find our replies below in red.

The paper describes an experiment to monitor numerous small lakes in the PPR region in USA, which expand/shrink and increase/decrease in number according to precipitations.

The authors used time series of the Copernicus Sentinel-1 SAR mission, together with a high-resolution DTM, to detect the extension of water on each of the time series images, which in this area reach a temporal sampling rate of 12 days.

The paper appears well written and describes a great deal of work. The choice of data and processing is effective, and the conclusions appear convincing and interesting. I particularly appreciate the use of the VH polarization channel as an additional information source, as it appears somewhat underrated in the current literature.

The processing algorithm is based on a Bayesian framework to derive maps of posterior probabilities that each pixel is inundated in each time slice, relying on priors and, in this case, external ancillary data (the DTM) for the a priori water and non-water distributions. The Bayesian framework allows to derive such maps as real numbers (the actual probability value), while the authors in the end perform some thresholding to arrive at binary water/non water maps. Although clearly this representation is more straightforward to interpret by a wider audience, I find that a good deal of information is actually lost with the thresholding. Dealing with a continuous indicator such as the final water probability allows to retain some form of confidence measure about the presence of water given the other sources of information, which is somewhat more complete than a binary map.

Moreover, it allows using a different set of evaluation tools. For instance, ROC analysis allows to derive detection efficiencies (e.g. the AUC), as well as threshold values to binarize the maps, optimized with respect to the different pixel populations. I would like to know the authors' opinion on this issue, and at least see some comments in the manuscript justifying their choices.

This is a very helpful comment. It is true that information is lost when binarizing the probability maps. In the end we chose to binarize them in order to make it easier to compute the water areas reported in section 3.2 as this was our main interest in this paper. The information on water probability is however useful.

The two maps shown in Fig. R2 show the posterior probabilities, $p(w|\sigma^0_{VH})$ and $p(w|\sigma^0_{VV})$, for VH (in green) and VV (in blue) polarisation, respectively, for the binary maps shown in Figure 6c,d) of the manuscript. As it can be seen, the borders between high and low water probability are quite crisp in most of the cases. This is due to the inclusion of the HAND-derived $p(W)$ prior probability. There are some areas with intermediate p values, mainly between high-probability wetlands.

This is obviously important for illustrating our conclusion that the integration of HAND helps with the avoidance of false positives. We added this figure to Appendix B of the revised manuscript and some considerations in lines 370-375 of the revised manuscript.

[Figure]

Fig. R2: Maps of posterior probabilities, $p(w/\sigma^0_p)$, for $p$ = VH (in green) and $p$ = VV (in blue) polarisations. a) corresponds to the binary maps shown in Fig. 6c) and b) to Fig. 6d).

Apart from this "methodological" point, I have only some minor comments as follows.

Line 226: it would be useful to clarify what are these two "histogram portions"... maybe add an equation (as for eqs. 5 and 6)...?

The term "histogram portions" is not well chosen. We changed this sentence to "where $\mu_1$ is the mean of all $\sigma^0_p \leq \tau_{ip}$ and $\mu_2$ the mean of all $\sigma^0_p > \tau_{ip}$. $\sigma_1^2$, $\sigma_2^2$ are their variances." (line 229 of the revised manuscript). $\tau_i$ is the initial Otsu threshold and is defined in line 226 of the revised manuscript.

Line 236: "where... respectively". This sentence is not very clear to me.

The word "respectively" is used erroneously here, we removed it. In addition, we changed line 235 (revised version) to "The parameters of the backscatter distributions for water, w, and land, l, for each pothole were estimated as …"

Line 246: this mention of Bayes' theorem could be moved somewhere before eq. (4) for clarity.

We propose to mention it in line 114 to have it in the introduction so that the reader knows what to expect: "We use a probabilistic approach based on Bayes' theorem combining SAR backscatter…"

More generally, the procedure described in sect. 2.3 is a little bit involved, so it would be beneficial to add a flow-chart or a pseudo-code algorithm description to better clarify how it works.

Figure 3 shows a flowchart of the overall procedure but it is not well bound into the text. We added some details on the "pothole specific threshold retrieval" and numbered the key elements of the flowchart. The different elements are now referred to in the text and it should be clearer for the reader which step of the overall processing is being described.

Figure 6: labels c) and d) are not visible (black background). More generally, labels should be probably enlarged a little for visibility. How are the false color composites in panels e) and f) obtained exactly?

Thank you for pointing this out. We changed the label colour and enlarged the labelling of the map coordinates. We also modified the caption (also for Fig. A1): "false-colour composites (red – near infrared, green – red, blue – green) of NAIP imagery acquired 4 July 2016."

Line 394: "adverse" -> "opposite"?

We changed it.

Figure 8: please add some explanation and possibly a reference for the LOESS acronym and meaning.

LOESS stands for "locally estimated scatterplot smoothing" and is described in Cleveland et al. (1992). It is also often referred to as the Savitzky-Golay filter. We added some explanation in the text (line 291-292 of the revised manuscript).

Line 409: correct citation.

We will correct it.

Lines 407-416: I believe this period contains points which are partially repeated later in the following paragraph. You may want to consider merging this with the subsequent text.

In response to the comments of another reviewer we proposed to reorganise section 3.2. By having a clearer structure it will become easier to avoid repetitions such as these.

In the revised version we split the section into two subsections: in 3.2.1, we describe the results by explaining the intra-annual dynamics of number of water bodies, total water area and median water extent as observed using Sentinel-1 data (Figures 8-9). In 3.2.2, we describe inter-annual variations and discuss them in the context of hydrometeorological conditions (i.e. discharge, precipitation, PDSI). This structure should be much clearer.

References:

W. S. Cleveland, E. Grosse and W. M. Shyu (1992) Local regression models. Chapter 8 of Statistical Models in S eds J.M. Chambers and T.J. Hastie, Wadsworth & Brooks/Cole.

**Reviewer #4**

We would like to thank the anonymous reviewer for the detailed and helpful comments. Please find our replies below in red.

The manuscript entitled "Monitoring surface water dynamics in the Prairie Pothole region using Dual-Polarized Sentinel-1 SAR time series" by Schlaffer et al. developed a new approach for classifying open water extent dynamic. The manuscript is novel and well organized. Most sections are informative, which can also lead to a long description and thus need to be significantly improved before acceptance. In the revised version, authors need to revise the writing style of this manuscript.

1. In this study, authors used many data sources to achieve their goals and these data are not in the same spatial resolution. For instance, the spatial resolution of Sentinel-1 data, the digital terrain model, and land cover are 20 m, 1m, and 30 m, respectively. To upscale the resolution from lower to higher is fine, while to downscale from lower to higher resolution can introduce uncertainty in data analysis, particularly for this study. In this study, land cover data (CDL) in 2015, which scaled from 30 m to 10 m, are used as reference data. Such a downscale can introduce potential uncertainty and should be discussed in the manuscript.

   In this case, CDL was used to sample pixels in order to characterise backscatter distributions for water and land areas as well as to draw samples of HAND values for fitting equation 7. From the CDL classes only the water class was of interest to us, whereas the rest was combined to a non-water class. In both cases, pixels at the border between water and land areas were masked in order to avoid sampling from mixed pixels. Masking was done in both directions, towards water and land, i.e. a distance of 20 m along the water-land border was avoided. This was already mentioned in line 219 of the original manuscript for the characterisation of backscatter distributions but not for the HAND sampling. In order to avoid repetition, we now described the masking in section 2.2.3 along with the other pre-processing (see line 180-184 of the revised manuscript).

2. In this manuscript, for Eqn. (7), the author only included the regression model. However, for data transparency, authors should list the regression parameter in the supplementary material.

   We added the used regression parameters $b_0$ = 1.9479, $b_1$ = -3.5598 to the caption of Fig. 5.

3. 3.For section 3.2 surface water dynamic, the authors described their results based on the temporal dynamics, i.e., only describe the results change with time, which helps understand model accuracy. At the same time, the authors compared surface water dynamic with wet or

dry conditions. In sum, this section is too long and not well-organized. I suggest authors can describe the temporal dynamics in the first section while discussing how surface dynamics change with hydrological conditions (wet or dry conditions). In this way, the readers can easily get the ideas that you want to express.

This is a very helpful comment. We propose to reorganise section 3.2 by splitting the section into two subsections: in 3.2.1, we describe the results for intra-annual variations by explaining the seasonality of number of water bodies, total water area and median water extent as observed using Sentinel-1 data (Figures 8-9). In 3.2.2, we first describe the inter-annual variations and then discuss our observations in the context of hydrometeorological conditions (i.e. discharge, precipitation, dry and wet periods as indicated by PDSI). This structure should be much clearer.

4. Research title is not specific. The current title can mislead readers that authors had mapped surface water dynamics in the whole prairie pothole region. Actually, this is a case study. Please revise your research title.

We changed the title to "Monitoring Surface Water Dynamics in the Prairie Pothole Region of North Dakota Using Dual-Polarised Sentinel-1 SAR Time Series".

5. The abstract and introduction are informative, but they are too long. For instance, the authors discussed the many key points, I can easily lose while I am reading the paper. I strongly suggest authors shorten the abstract and introduction. In addition, the conclusion section is well-written. Please follow this style to revise your abstract.

We shortened the abstract from ca. 480 to 380 words:

"The North American Prairie Pothole Region (PPR) represents a large system of wetlands with great importance for biodiversity, water storage and flood management. Knowledge of seasonal and inter-annual surface water dynamics in the PPR is important for understanding the functionality of these wetland ecosystems and the changing degree of hydrologic connectivity between them. Optical sensors widely used for retrieving such information are often limited by their temporal resolution and cloud cover, especially in the case of flood events. Synthetic aperture radar (SAR) sensors can potentially overcome such limitations. However, water extent retrieval from SAR data is often impacted by environmental factors, such as wind on water surfaces. Hence, for reliably monitoring water extent over longer time periods robust retrieval methods are required.

The aim of this study was to develop a robust approach for classifying open water extent in the PPR and to analyse the obtained time series covering the entire available Sentinel-1 observation period from 2015 to 2020 in the hydrometeorological context. Open water in prairie potholes was classified by fusing dual-polarised Sentinel-1 data and high-resolution topographical information using a Bayesian framework. The approach was tested for a study area in North Dakota. The resulting surface water maps were validated using high-resolution airborne optical imagery. For the observation period, the total water area, the number of water bodies and the median area per water body were computed. The validation of the retrieved water maps yielded producer's accuracies between 84 % and 95 % for calm days and between 74 % and 88 % on windy days. User's accuracies were above 98 % in all cases, indicating a very low occurrence of false positives due to the constraints introduced by topographical information.

The observed dynamics of total water area displayed both intra-annual and inter-annual patterns. In addition to differences in seasonality between small (< 1 ha) and large (> 1 ha) water bodies due to the effect of evaporation during summer, these size classes also responded differently to an extremely wet period from 2019 to 2020 in terms of increase in number and total area covered. The results demonstrate the potential of Sentinel-1 data for high-resolution monitoring of prairie wetlands. Limitations exist related to wind inhibiting correct water extent retrieval and due to the rather long acquisition interval of 12 days over the PPR."

As for the introduction, given the substantive amount of work which has been dedicated to the PPR over the last few years in the field of remote sensing we see the necessity to describe the other technologies used for remote sensing of wetlands in the PPR (optical satellite and airborne imagery) and make a point for the value of Sentinel-1 SAR data in addition to optical satellite/airborne and, e.g., Radarsat-2 SAR data. We also have to describe the most important modelling studies (e.g., Liu and Schwartz, 2011) as they are important for the understanding of the findings. Therefore, we feel we cannot substantially shorten the introduction.

6. In figures 8 and 9, the authors only compared the water body with wet or dry seasons. I recommend authors can plot the hydrography on figures 8 and 9. Authors can compare the relationship between flood and drought conditions. In this way, the relationship between hydrology and water bodies can be quantitively assessed. Authors can relate the hydrology data to the surface water body.

In the text, we discuss the observed surface water dynamics in the context of wet and dry periods and refer to the PDSI time series shown in Figure 2. PDSI is already used when introducing the study site and hydrological/meteorological events during the study period (section 2.1). It would be possible to show PDSI in figure 8 instead, however, the data are already very useful in section 2.1. We hope that the restructuring of section 3.2 in response to your previous comment clarifies the relationships between inter-annual surface water extent and hydrometeorological conditions.

7. Line 89, please list the references here.

Thank you for pointing this out. We added the references.

8. Here are some thoughts for authors' reference. 1) In this study, authors used a high-resolution digital terrain model (1m). For the topography, there are many topographic indices, for instance, topographic wetness index (TWI) and others, which can represent the topographic information in detail. In my thoughts, the appearance of surface water is highly related to topography information. I suggest authors can try to relate water body location with TWI or other indices. This may help authors to validate your results. 2) Authors extracted the water body from Sentinel-1 data. To date, Google Earth has spatial resolution data. In authors' data, authors validated the results using the same data source but for different locations. The validation is good. However, this data is not easy to obtain. Can you try to validate the data through Google Earth. I know this is beyond the scope of this study, but authors can try this in future studies.

Thank you for these interesting suggestions.

1) In this case we used Height Above Nearest Drainage (HAND) to represent the topographic information in relation to the drainage network. Of course, TWI may yield additional information and could be used as a predictor variable in Eq. 7 instead or in addition to HAND.

2) In this case we used imagery from the National Agriculture Imagery Program (NAIP) which we downloaded from USGS Earth Explorer. We are aware that Google Earth Engine (GEE) also contains NAIP imagery. However, we found it simpler to download the data and take the reference samples locally.

References:

Liu, G. and Schwartz, F. W.: An integrated observational and model-based analysis of the hydrologic response of prairie pothole systems to variability in climate, Water Resources Research, 47, 1–15, https://doi.org/10.1029/2010WR009084, 2011.

**List of most important changes (all line numbers refer to the track-changes version of the revised manuscript):**

| Line numbers (track-changes) | Change |
|---|---|
| 1-32 | Shortened the abstract from 480 to 380 words |
| 41-44 | Made introduction of PPR more descriptive (moved a sentence from l. 135) |
| 95-99 | Added better explanation of cross- and co-polarisation |
| 122 | Mentioned the use of Bayes' theorem in the study |
| Figure 1 | Added north arrow and footprints of NAIP images |
| 177-182 | More extensive description of HAND algorithm |
| 191-195 | Description of masking of land-cover dataset pixels along land-water borders |
| Figure 3 | Added detail to the "pothole specific analysis" and added numbers to most important processing steps (referred to in lines 219, 228, 265, 284) |
| 238-243 | Introduced initial threshold and added better explanation of symbols in eq. (1) |
| 304-305 | Added explanation and reference for LOESS smoothing |
| Figure 5 | Added north arrow and regression parameters to the caption. Added explanation of UTM |
| 340, 531 | Changed dual-pol to dual-polarised |
| Figure 6 | Changed letters c and d to white, enlarged map coordinates, added explanation of false-colour compositing to the caption. Added north arrow to panel b |
| Figure 7 | Added north arrow to panel b |
| 370-375 | Added a paragraph discussing the probabilistic maps in Fig. B1 and the reason for binarising the maps. |
| 376-512 | Restructured section 3.2 as follows: 3.2 Surface water dynamics (introductory description of Fig. 8, discussion of quality of some data points) 3.2.1 Intra-annual dynamics |

| | 3.2.2 Inter-annual dynamics |
|---|---|
| 412-419 | Moved to 3.2.2 |
| 445-453 | Moved to 3.2.1 |
| 473-475 | Removed as this was unclear |
| 479-488 | This was partly moved to 3.2.1 and partly removed as these aspects were already covered earlier on (remark by reviewer #3) |
| 505-512 | Merged with existing text in lines 470-478 |
| Figure 10 | Added a zoom in to a sub-window (panels b,d); darker background for higher contrast |
| Figure A1 | Added explanation of false-colour compositing to the caption |
| Figure B1 | Added maps of posterior water probability |

---

## Author Response (AR2)

**HESS-2021-330, Revision 2, Authors' response**

Dear Zhongbo,

in addition to the changes made during the first round of revisions, we completed a thorough editing of the manuscript.

This includes increasing the font sizes in Figures 4 and 7 (labels of the other Figures were already fixed in the first round). Additionally, we fixed some small details in the text and added acknowledgements to the four reviewers and financial support for publication costs.

Changes made in the first round of revision are marked in blue, changes made during the second round are marked in orange in the track-changes version of the manuscript.
An automatically produced, detailed list of changes can be found on the last pages of the track-changes version. "r2" marks changes made during the second round.

Kind regards,

The authors